# Exploitability Minimization in Games and Beyond

**Denizalp Goktas**
Department of Computer Science
Brown University
Providence, RI 02906, USA
denizalp_goktas@brown.edu

**Amy Greenwald**
Brown University
Providence, RI 02906, USA
amy_greenwald@brown.edu

## Abstract

Pseudo-games are a natural and well-known generalization of normal-form games, in which the actions taken by each player affect not only the other players' payoffs, as in games, but also the other players' strategy sets. The solution concept par excellence for pseudo-games is the generalized Nash equilibrium (GNE), i.e., a strategy profile at which each player's strategy is feasible and no player can improve their payoffs by unilaterally deviating to another strategy in the strategy set determined by the other players' strategies. The computation of GNE in pseudo-games has long been a problem of interest, due to applications in a wide variety of fields, from environmental protection to logistics to telecommunications. Although computing GNE is PPAD-hard in general, it is still of interest to try to compute them in restricted classes of pseudo-games. One approach is to search for a strategy profile that minimizes exploitability, i.e., the sum of the regrets across all players. As exploitability is nondifferentiable in general, developing efficient first-order methods that minimize it might not seem possible at first glance. We observe, however, that the exploitability-minimization problem can be recast as a min-max optimization problem, and thereby obtain polynomial-time first-order methods to compute a refinement of GNE, namely the variational equilibria (VE), in convex-concave cumulative regret pseudo-games with jointly convex constraints. More generally, we also show that our methods find the stationary points of the exploitability in polynomial time in Lipschitz-smooth pseudo-games with jointly convex constraints. Finally, we demonstrate in experiments that our methods not only outperform known algorithms, but that even in pseudo-games where they are not guaranteed to converge to a GNE, they may do so nonetheless, with proper initialization.

## 1   Introduction

Nash equilibrium [1] is the canonical solution concept used to predict the outcome of models of rational agents known as games. A **game** comprises a set of players, each of whom simultaneously chooses a strategy (or action) from a strategy set, and then receives a payoff based on all the players' choices. A **Nash equilibrium (NE)** is a strategy profile, i.e., a collection of strategies, one per player, from which no player can improve their payoff via a unilateral deviation.

Recently, the literature on NE has expanded beyond classic economic applications to the question of its computation, a problem known as the **Nash equilibrium problem (NEP)** [2, 3]. Unfortunately, it is unlikely that there exists an efficient algorithm to compute NE, as NEP is PPAD-complete, i.e., NEP is equivalent to the problem of computing solutions to a class of fixed-point problems known as PPAD [4], which are believed to be computationally intractable [5, 6, 7]. Nonetheless, there has been a great deal of recent progress developing *efficient* algorithms to compute NE in special classes of games, such as two-player zero-sum [8, 9, 10, 11, 12, 13, 14, 15, 16, 17, 18, 19, 20, 21, 22, 23, 24, 25,

36th Conference on Neural Information Processing Systems (NeurIPS 2022).

26, 27, 28, 29, 30, 31], potential [32, 33, 34], and monotone (or, more generally, variationally stable) games [35, 36, 37, 38, 39, 40, 41]. Interest in two-player zero-sum games, for example, stems in part from emerging applications in machine learning, e.g., the training of GANs [42, 43].

A natural generalization of games are **pseudo-games** or **abstract economies**, first introduced by Arrow and Debreu [44] in the context of markets, in which the actions taken by each player affect not only the other players' payoffs, as in games, but also the other players' strategy sets. In other words, players (simultaneously) take actions that can restrict the other players' strategy sets. Such a model is not technically a game, giving rise to the pseudo-game terminology. Pseudo-games generalize games, and hence are even more widely applicable. Some recently studied applications include adversarial classification [45, 46], energy resource allocation [47, 48], environmental protection [49, 50], cloud computing [51, 52], adversarial classification, ride sharing services [53], transportation [54], wireless and network communication [55, 56].

The solution concept par excellence for pseudo-games is the **generalized Nash equilibrium (GNE)** [44, 57, 58], i.e., a strategy profile at which each player's strategy is feasible and no player can improve their payoffs by unilaterally deviating to another strategy in the strategy set determined by the other players' strategies. The aforementioned applications speak to the importance of developing algorithms that can approximate GNEs efficiently in the largest class of pseudo-games possible. Work in this direction, on the **generalized Nash equilibrium problem (GNEP)**, is progressing; see, for example, [59, 57, 60, 61, 62, 63, 64, 65, 66, 67, 68, 69, 70, 71, 72, 73, 74, 75, 76, 77]. Nonetheless, there are still few, if any [78], GNE-finding algorithms with computational guarantees, even for restricted classes of pseudo-games.

**Exploitability**, or the Nikoida-Isoda potential [79], is defined as the sum of the players' payoff-maximizing unilateral deviations w.r.t. a given strategy profile. Minimizing exploitability is a logical approach to computing GNE, especially in cases where exploitability is convex, such as in the pseudo-games that result from replacing each player's payoff function in a monotone pseudo-game by a second-order Taylor approximation [80]. When the players' payoffs are also Lipschitz-smooth and strongly concave (in their own action), this approximation can be quite good, because the error in the approximation will be bounded by the Lipschitz-smoothness and strong concavity constant. *Minimizing exploitability can thus be an effective way of finding GNEs in pseudo-games; however, to our knowledge, no convergence rates for this approach are known, and few methods have been proposed for pseudo-games whose exploitability is non-convex.*

In this paper, we develop efficient exploitability-minimization algorithms that converge to variational equilibrium (VE), a refinement of GNE, exactly in a large class of pseudo-games with jointly convex constraints (which includes, but is not limited to, two-player zero-sum, $n$-player pairwise zero-sum, and a large class of monotone and bilinear pseudo-games, as well as Cournot oligopoly games); and approximately, in Lipschitz-smooth pseudo-games with jointly convex constraints, where exact computation is intractable. Our algorithms apply not only to games with discrete action spaces but also to games with continuous action spaces; previous exploitability-minimization algorithms concern only finite games [81, 82].

The **cumulative regret** of two strategy profiles is defined as the sum, across all players, of the change in utility due to unilateral deviations from one strategy profile to the other. Our algorithms and their analysis rely on the following key insights: assuming jointly convex constraints (i.e., symmetric and convex-valued joint constraint correspondences) the exploitability-minimization problem is equivalent to solving a cumulative regret min-max optimization problem, which although non-convex-concave in general can nonetheless be solved by **gradient descent ascent (GDA)** methods[1] [25, 85] using a parametric variant of Moreau's envelope theorem that we introduce (Theorem 4.1). We apply this logic to develop two algorithms for Lipschitz-smooth pseudo-games with jointly convex constraints. *To the best of our knowledge, this makes ours the first exploitability-minimization methods with convergence rate guarantees in this class of pseudo-games.*

The first (EDA; Algorithm 1) is an extragradient method that converges in average iterates to an $\varepsilon$-VE, and hence an $\varepsilon$-GNE, in $O(1/\varepsilon)$ iterations in pseudo-games with convex-concave cumulative regret, where $\varepsilon$ is the exploitability of the game (Theorem 3.4). This class of pseudo-games includes zero-sum, potential, Cournot, a large class of monotone pseudo-games, and a class of bilinear

---

[1]When the objective function is convex-concave, a saddle point is guaranteed to exist, in which case this optimization problem can be solved using simultaneous-GDA-style algorithms (e.g., [21, 83, 84]).

pseudo-games, namely those with convex second-order approximations. Our second algorithm (ADA; Algorithm 2) is a nested GDA algorithm, whose best iterate converges more generally to a stationary point of (regularized) exploitability in all Lipschitz-smooth pseudo-games with jointly convex constraints at the slightly slower rate of $O(\log(1/\varepsilon)/\varepsilon)$ (Theorem 4.2). When the (regularized) exploitability is additionally invex, our method also converges to a GNE in best iterates at the same rate. This convergence can be strengthened to convergence in last iterates in $O(\log(1/\varepsilon)^2)$ iterations in strongly-convex exploitability pseudo-games and in a certain class of pseudo-games with non-convex exploitability (Theorem 4.3).

Pseudo-games with jointly convex constraints are of interest in decentralized resource allocation problems with feasibility (e.g., supply) constraints. Feasibility constraints are almost always joint, and often also jointly convex, since we generally require the sum of the allocations to be less than or equal to the supply. One of the many examples of such pseudo-games is wireless and network communication, where agents communicate packets across a shared network, with limited capacities [55]. The experiments we run in this paper suggest that our algorithms have the potential to yield improved solutions to such pseudo-games as compared to other known approaches, and thus open up avenues for practical applications of our work in the future.

**Related Work**    Following Arrow and Debreu's introduction of GNE, Rosen [86] initiated the study of the mathematical and computational properties of GNE in pseudo-games with jointly convex constraints, proposing a projected gradient method to compute GNE. Thirty years later, Uryas'ev and Rubinstein [87] developed the first relaxation methods for finding GNEs, which were improved upon in subsequent works [88, 89]. Two other types of algorithms were also introduced to the literature: Newton-style methods [59, 64, 68, 69, 70, 90] and interior-point potential methods [90]. Many of these approaches are based on minimizing the exploitability of the pseudo-game, but others use variational inequality[91, 92] and Lemke methods [93].

More recently, novel methods that transform GNEP to NEP were analyzed. These models take the form of either exact penalization methods, which lift the constraints into the objective function via a penalty term [72, 73, 76, 77, 94], or augmented Lagrangian methods [71, 74, 76, 95], which do the same, augmented by dual Lagrangian variables. Using these methods, Jordan, Lin, and Zampetakis [78] provide the first convergence rates to a $\varepsilon$-GNE in monotone (resp. strongly monotone) pseudo-games with jointly affine constraints in $\tilde{O}(1/\varepsilon)$ $(\tilde{O}(1/\sqrt{\varepsilon}))$ iterations. These algorithms, despite being highly efficient in theory, are numerically unstable in practice [78]. Nearly all of the aforementioned approaches concerned pseudo-games with jointly convex constraints.

Exploitability minimization has also been a valuable tool in multi-agent reinforcement learning; algorithms in this literature that aim to minimize exploitability are known as exploitability descent algorithms [81]. Lockhart et al. [81] analyzed exploitability descent in two-player, zero-sum, extensive-form games with finite action spaces. Variants of exploitability-descent have also been combined with entropic regularization and homotopy methods to solve for NE in large games [82].

## 2    Preliminaries

**Notation**    We use caligraphic uppercase letters to denote sets (e.g., $\mathcal{X}$); bold lowercase letters to denote vectors (e.g., $\boldsymbol{p}, \boldsymbol{\pi}$); bold uppercase letters to denote matrices (e.g., $\boldsymbol{X}$) and lowercase letters to denote scalar quantities (e.g., $x, \delta$). We denote the $i$th row vector of a matrix (e.g., $\boldsymbol{X}$) by the corresponding bold lowercase letter with subscript $i$ (e.g., $\boldsymbol{x}_i$). Similarly, we denote the $j$th entry of a vector (e.g., $\boldsymbol{p}$ or $\boldsymbol{x}_i$) by the corresponding Roman lowercase letter with subscript $j$ (e.g., $p_j$ or $x_{ij}$). We denote functions by a letter determined by the value of the function, e.g., $f$ if the mapping is scalar-valued, $\boldsymbol{f}$ if the mapping is vector-valued, and $\mathcal{F}$ if the mapping is set-valued. We denote the set of integers $\{1, \ldots, n\}$ by $[n]$, the set of natural numbers by $\mathbb{N}$, the set of real numbers by $\mathbb{R}$, and the positive and strictly positive elements of a set by a $+$ and $++$ subscript, respectively, e.g., $\mathbb{R}_+$ and $\mathbb{R}_{++}$. We define the gradient $\nabla_{\boldsymbol{x}} : C^1(\mathcal{X}) \to C^0(\mathcal{X})$ as the operator which takes as input a functional $f : \mathcal{X} \to \mathbb{R}$, and outputs a vector-valued function consisting of the partial derivatives of $f$ w.r.t. $\boldsymbol{x}$. We denote the orthogonal projection onto a set $C$ by $\Pi_C$, i.e., $\Pi_C(\boldsymbol{x}) = \arg\min_{\boldsymbol{y} \in C} \|\boldsymbol{x} - \boldsymbol{y}\|_2^2$. Finally, we write $\mathbb{1}_{\mathcal{X}}(\boldsymbol{a})$ to denote the indicator function of the set $\mathcal{X}$, i.e., $\mathbb{1}_{\mathcal{X}}(\boldsymbol{x}) = 0$ if $\boldsymbol{x} \in \mathcal{X}$ and $\mathbb{1}_{\mathcal{X}}(\boldsymbol{x}) = \infty$ if $\boldsymbol{x} \notin \mathcal{X}$.

A **pseudo-game** [44] $\mathcal{G} \doteq (n, \mathcal{A}, \mathcal{X}, \boldsymbol{h}, \boldsymbol{u})$ comprises $n \in \mathbb{N}_+$ players, each $i \in [n]$ of whom chooses an $\boldsymbol{a}_i$ from an action space $\mathcal{A}_i$. We denote the players' joint action space by $\mathcal{A} = \bigtimes_{i \in [n]} \mathcal{A}_i$. Each player $i$ aims to maximize their continuous utility $u_i : \mathcal{A} \to \mathbb{R}$, which is concave in $\boldsymbol{a}_i$, by choosing a feasible action from a set of actions $\mathcal{X}_i(\boldsymbol{a}_{-i}) \subseteq \mathcal{A}_i$ determined by the actions $\boldsymbol{a}_{-i} \in \mathcal{A}_{-i} \subset \mathbb{R}^m$ of the other players, where $\mathcal{X}_i : \mathcal{A}_{-i} \rightrightarrows \mathcal{A}_i$ is a non-empty, continuous, compact- and convex-valued action correspondence, which for convenience we represent as $\mathcal{X}_i(\boldsymbol{a}_{-i}) = \{\boldsymbol{a}_i \in \mathcal{A}_i \mid h_{ik}(\boldsymbol{a}_i, \boldsymbol{a}_{-i}) \geq \boldsymbol{0}, \text{ for all } k \in [d]\}$, where for all $i \in [n]$, and $k \in [d]$, $h_{ik}$ is a continuous and concave function in $\boldsymbol{a}_i$, which defines the constraints. We denote the product of the feasible action correspondences of all players by $\mathcal{X}(\boldsymbol{a}) = \bigtimes_{i \in [n]} \mathcal{X}_i(\boldsymbol{a}_{-i})$, which we note is guaranteed to be non-empty, continuous, and compact-valued, but not necessarily convex-valued. Additionally, overloading notation, we define the set of jointly feasible strategy profiles $\mathcal{X} = \{\boldsymbol{a} \in \mathcal{A} \mid h_{ik}(\boldsymbol{a}) \geq \boldsymbol{0}, \text{ for all } i \in [n], k \in [d]\}$.

A two player pseudo-game is called **zero-sum** if for all $\boldsymbol{a} \in \mathcal{A}$, $u_1(\boldsymbol{a}) = -u_2(\boldsymbol{a})$. A pseudo-game is called **monotone** if for all $\boldsymbol{a}, \boldsymbol{b} \in \mathcal{A}$ $\sum_{i \in [n]} \left(\nabla_{\boldsymbol{a}_i} u_i(\boldsymbol{a}) - \nabla_{\boldsymbol{a}_i} u_i(\boldsymbol{b})\right)^T (\boldsymbol{a} - \boldsymbol{b}) \leq 0$. A pseudo-game is said to have **joint constraints** if there exists a function $\boldsymbol{g} : \mathcal{A} \to \mathbb{R}^d$ s.t. $\boldsymbol{g} = \boldsymbol{h}_1 = \ldots = \boldsymbol{h}_d$. If additionally, for all $k \in [d]$, $g_k(\boldsymbol{a})$ is concave in $\boldsymbol{a}$, then we say that the pseudo-game has **jointly convex constraints**, as this assumption implies that $\mathcal{X}$ is a convex set. A **game** [1] is a pseudo-game where, for all players $i \in [n]$, $\mathcal{X}_i$ is a constant correspondence, i.e., for all players $i \in [n]$, $\mathcal{X}_i(\boldsymbol{a}_{-i}) = \mathcal{X}_i(\boldsymbol{b}_{-i})$, for all $\boldsymbol{a}, \boldsymbol{b} \in \mathcal{A}$. Moreover, while $\mathcal{X}(\boldsymbol{a})$ is convex, for all action profiles $\boldsymbol{a} \in \mathcal{A}$, $\mathcal{X}$ is not guaranteed to be convex unless one assumes joint convexity.

Given a pseudo-game $\mathcal{G}$, an $\varepsilon$-**generalized Nash equilibrium (GNE)** is strategy profile $\boldsymbol{a}^* \in \mathcal{X}(\boldsymbol{a}^*)$ s.t. for all $i \in [n]$ and $\boldsymbol{a}_i \in \mathcal{X}_i(\boldsymbol{a}^*_{-i})$, $u_i(\boldsymbol{a}^*) \geq u_i(\boldsymbol{a}_i, \boldsymbol{a}^*_{-i}) - \varepsilon$. An $\varepsilon$-**variational equilibrium (VE)** (or **normalized GNE**) of a pseudo-game is a strategy profile $\boldsymbol{a}^* \in \mathcal{X}(\boldsymbol{a}^*)$ s.t. for all $i \in [n]$ and $\boldsymbol{a} \in \mathcal{X}$, $u_i(\boldsymbol{a}^*) \geq u_i(\boldsymbol{a}_i, \boldsymbol{a}^*_{-i}) - \varepsilon$. We note that in the above definitions, one could just as well write $\boldsymbol{a}^* \in \mathcal{X}(\boldsymbol{a}^*)$ as $\boldsymbol{a}^* \in \mathcal{X}$, as any fixed point of the joint action correspondence is also a jointly feasible action profile, and vice versa. A GNE (VE) is an $\varepsilon$-GNE (VE) with $\varepsilon = 0$. Under our assumptions, while GNE are guaranteed to exist in all pseudo-games by Arrow and Debreu's lemma on abstract economies [44], VE are only guaranteed to exist in pseudo-games with jointly convex constraints [65]. Note that the set of $\varepsilon$-VE of a pseudo-game is a subset of the set of the $\varepsilon$-GNE, as $\mathcal{X}(\boldsymbol{a}^*) \subseteq \mathcal{X}$, for all $\boldsymbol{a}^*$ which are GNE of $\mathcal{G}$. The converse, however, is not true, unless $\mathcal{A} \subseteq \mathcal{X}$. Further, when $\mathcal{G}$ is a game, GNE and VE coincide; we refer to this set simply as NE.

**Mathematical Preliminaries**  Finally, we define several mathematical concepts that are relevant to our convergence proofs. Given $A \subset \mathbb{R}^n$, the function $f : \mathcal{A} \to \mathbb{R}$ is said to be $\ell_f$-**Lipschitz-continuous** iff $\forall \boldsymbol{x}_1, \boldsymbol{x}_2 \in \mathcal{X}, \|f(\boldsymbol{x}_1) - f(\boldsymbol{x}_2)\| \leq \ell_f \|\boldsymbol{x}_1 - \boldsymbol{x}_2\|$. If the gradient of $f$ is $\ell_{\nabla f}$-Lipschitz-continuous, we then refer to $f$ as $\ell_{\nabla f}$-**Lipschitz-smooth**. A function $f : \mathcal{X} \to \mathbb{R}$ is said to be **invex** w.r.t. a function $\boldsymbol{h} : \mathcal{X} \times \mathcal{X} \to \mathcal{X}$ if $f(\boldsymbol{x}) - f(\boldsymbol{y}) \geq \boldsymbol{h}(\boldsymbol{x}, \boldsymbol{y}) \cdot \nabla f(\boldsymbol{y})$, for all $\boldsymbol{x}, \boldsymbol{y} \in \mathcal{X}$. A function $f : \mathcal{X} \to \mathbb{R}$ is **convex** if it is invex w.r.t. $\boldsymbol{h}(\boldsymbol{x}, \boldsymbol{y}) = \boldsymbol{x} - \boldsymbol{y}$ and **concave** if $-f$ is convex. A function $f : \mathcal{X} \to \mathbb{R}$ is $\mu$-**strongly convex (SC)** if $f(\boldsymbol{x}_1) \geq f(\boldsymbol{x}_2) + \langle \nabla_{\boldsymbol{x}} f(\boldsymbol{x}_2), \boldsymbol{x}_1 - \boldsymbol{x}_2 \rangle + \mu/2 \|\boldsymbol{x}_1 - \boldsymbol{x}_1\|^2$, and $\mu$-**strongly concave** if $-f$ is $\mu$-strongly convex.

## 3  From Exploitability Minimization To Min-Max Optimization

In this section, we reformulate the exploitability minimization problem as a min-max optimization problem. This reformulation is a consequence of the following observation, which is key to our analysis: *In pseudo-games for which a VE exists, e.g., pseudo-games with jointly convex constraints, the quasi-optimization problem  of minimizing exploitability, whose solutions characterize the set of GNEs, reduces to the reduces to a standard min-max optimization problem (with independent constraint sets), which characterizes the set of VEs, a subset of the GNEs.* This new perspective allows us to efficiently solve GNEP by computing VEs in a large class of pseudo-games as an immediate consequence of known results on the computation of saddle points in convex-concave min-max optimization problems.

Given a pseudo-game $\mathcal{G}$, we define the **regret** felt by any player $i \in [n]$ for an action $\boldsymbol{a}_i$ as compared to another action $\boldsymbol{b}_i$, given the action profile $\boldsymbol{a}_{-i}$ of other players, as follows: $\text{Regret}_i(\boldsymbol{a}_i, \boldsymbol{b}_i; \boldsymbol{a}_{-i}) = u_i(\boldsymbol{b}_i, \boldsymbol{a}_{-i}) - u_i(\boldsymbol{a}_i, \boldsymbol{a}_{-i})$. Additionally, the **cumulative regret**, or the **Nikaido-Isoda function**, $\psi : \mathcal{A} \times \mathcal{A} \to \mathbb{R}$ between two action profiles $\boldsymbol{a} \in \mathcal{X}$ and $\boldsymbol{b} \in \mathcal{X}$

across all players in a pseudo-game is given by $\psi(\boldsymbol{a}, \boldsymbol{b}) = \sum_{i \in [n]} \text{Regret}_i(\boldsymbol{a}_i, \boldsymbol{b}_i; \boldsymbol{a}_{-i})$. Further, the **exploitability**, or **Nikaido-Isoda *potential*** function, $\varphi : \mathcal{A} \to \mathbb{R}$ of an action profile $\boldsymbol{a}$ is defined as $\varphi(\boldsymbol{a}) = \sum_{i \in [n]} \max_{\boldsymbol{b}_i \in \mathcal{X}_i(\boldsymbol{a}_{-i})} \text{Regret}_i(\boldsymbol{a}_i, \boldsymbol{b}_i; \boldsymbol{a}_{-i})$ [57]. Notice that the max is taken over $\mathcal{X}_i(\boldsymbol{a}_{-i})$, since a player can only deviate within the set of available strategies.

A well-known [57][2] result is that any unexploitable strategy profile in a pseudo-game is a GNE.

**Lemma 3.1.** *Given a pseudo-game $\mathcal{G}$, for all $\boldsymbol{a} \in \mathcal{A}$, $\varphi(\boldsymbol{a}) \geq 0$. Additionally, a strategy profile $\boldsymbol{a}^* \in \mathcal{X}(\boldsymbol{a}^*)$ is a GNE iff it achieves the lowerbound, i.e., $\varphi(\boldsymbol{a}^*) = 0$.*

This lemma tells us that we can reformulate GNEP as the quasi-optimization problem of minimizing exploitability, i.e., $\min_{\boldsymbol{a} \in \mathcal{X}(\boldsymbol{a})} \varphi(\boldsymbol{a})$. Here, "quasi" refers to the fact that a solution to this problem is both a minimizer of $\varphi$ and a fixed point $\boldsymbol{a}^*$ s.t. $\boldsymbol{a}^* \in \mathcal{X}(\boldsymbol{a}^*)$.[3] Despite this reformulation of GNEP in terms of exploitability, no exploitability-minimization algorithms with convergence rate guarantees are known. The unexploitability (!) of exploitability may be due to the fact that it is not Lipschitz-smooth. The key insight that allows us to obtain convergence guarantees is that we treat the GNE problem not as a quasi-minimization problem, but rather as a quasi-min-max optimization problem, namely $\min_{\boldsymbol{a} \in \mathcal{X}(\boldsymbol{a})} \varphi(\boldsymbol{a}) = \min_{\boldsymbol{a} \in \mathcal{X}(\boldsymbol{a})} \max_{\boldsymbol{b} \in \mathcal{X}(\boldsymbol{a})} \psi(\boldsymbol{a}, \boldsymbol{b})$.

**Observation 3.2.** *Given a pseudo-game $\mathcal{G}$, for all $\boldsymbol{a} \in \mathcal{A}$, $\varphi(\boldsymbol{a}) = \max_{\boldsymbol{b} \in \mathcal{X}(\boldsymbol{a})} \psi(\boldsymbol{a}, \boldsymbol{b})$.*

*Proof.* The per-player maximum operators can be pulled out of the sum, as the $i$th player's best action in hindsight is independent of the other players' best actions in hindsight, since the action profile $\boldsymbol{a}$ is fixed:

$$\varphi(\boldsymbol{a}) = \sum_{i \in [n]} \max_{\boldsymbol{b}_i \in \mathcal{X}_i(\boldsymbol{a}_{-i})} \text{Regret}_i(\boldsymbol{a}_i, \boldsymbol{b}_i; \boldsymbol{a}_{-i}) = \max_{\boldsymbol{b} \in \mathcal{X}(\boldsymbol{a})} \sum_{i \in [n]} \text{Regret}_i(\boldsymbol{a}_i, \boldsymbol{b}_i; \boldsymbol{a}_{-i}) = \max_{\boldsymbol{b} \in \mathcal{X}(\boldsymbol{a})} \psi(\boldsymbol{a}, \boldsymbol{b})$$

That is, the optimization problem in the $i$th summand is independent of the optimization problem in the other summands, e.g., for any $\boldsymbol{y} \in [0, 1/2]^2$, $\max_{x_1 \in [0,1]:x_1-y_1 \geq 0} \{x_1 y_1\} + \max_{x_2 \in [0,1]:x_2-y_2 \geq 0} \{x_2 y_2\} = \max_{\boldsymbol{x} \in [0,1]^2:\boldsymbol{x}-\boldsymbol{y} \geq \boldsymbol{0}} x_1 y_1 + x_2 y_2$. $\square$

If we restrict our attention to VEs, a subset of GNEs, VE exploitability—*hereafter exploitability for short*—is conveniently expressed as $\varphi(\boldsymbol{a}) = \max_{\boldsymbol{b} \in \mathcal{X}} \psi(\boldsymbol{a}, \boldsymbol{b})$. When a VE exists, e.g., in pseudo-games with jointly convex constraints, this exploitability is guaranteed to achieve the lower bound of 0 for some $\boldsymbol{a} \in \mathcal{X}$. In such cases, formulation of exploitability minimization as a quasi-min-max optimization problem reduces to a standard min-max optimization problem, namely $\min_{\boldsymbol{a} \in \mathcal{X}} \max_{\boldsymbol{b} \in \mathcal{X}} \psi(\boldsymbol{a}, \boldsymbol{b})$, which characterizes VEs.

This problem is well understood when $\psi$ is a convex-concave objective function [21, 83, 84, 96]. Furthermore, the cumulative regret $\psi$ is indeed convex-concave, i.e., convex in $\boldsymbol{a}$ and concave in $\boldsymbol{b}$, in many pseudo-games of interest: e.g., two-player zero-sum, $n$-player pairwise zero-sum, and a large class of monotone and bilinear pseudo-games, as well as Cournot oligopoly games.

Going forward, we restrict our attention to pseudo-games satisfying the following assumptions. Lipschitz-smoothness is a standard assumption in the convex optimization literature [97], while joint convexity is a standard assumption in the GNE literature [57]. Furthermore, these are weaker assumptions than ones made to obtain the few known results on convergence rates to GNEs [78].

**Assumption 3.3.** *The pseudo-game $\mathcal{G}$ has joint constraints $\boldsymbol{g} : \mathcal{A} \to \mathbb{R}^d$, and additionally, 1. (Lipschitz smoothness and concavity) for any player $i \in [n]$, their utility function $u_i$ is $\ell_{\nabla_{\boldsymbol{u}}}$-Lipschitz smooth; 2. (Joint Convexity) $\boldsymbol{g}$ is component-wise concave.*

Using the simple observation that every VE of a pseudo-game is the solution to a min-max optimization problem we introduce our first algorithm (EDA; Algorithm 1), an extragradient method [83]. The algorithm works by interleaving extragradient ascent and descent steps: at iteration $t$, given $\boldsymbol{a}^{(t)}$, it ascends on $\psi(\boldsymbol{a}^{(t)}, \cdot)$, thereby generating a better response $\boldsymbol{b}^{(t+1)}$, and then descends on $\psi(\cdot, \boldsymbol{b}^{(t+1)})$, thereby decreasing exploitability. We combine several known results about the

---

[2]yet hard to exploit(!)

[3]This problem could also be formulated as minimizing over the set of jointly feasible actions, i.e., $\min_{\boldsymbol{a} \in \mathcal{X}} \varphi(\boldsymbol{a})$; however, without assuming joint convexity, $\mathcal{X}$ is not necessarily convex.

---

**Algorithm 1** Extragradient descent ascent (EDA)

---

**Inputs:** $n, \boldsymbol{u}, \mathcal{X}, \boldsymbol{\eta}, T, \boldsymbol{a}^{(0)}, \boldsymbol{b}^{(0)}$
**Outputs:** $(\boldsymbol{a}^{(t)}, \boldsymbol{b}^{(t)})_{t=0}^{T}$

1: **for** $t = 0, \dots, T-1$ **do**
2: $\quad \boldsymbol{a}^{(t+1/2)} = \Pi_{\mathcal{X}} \left[ \boldsymbol{a}^{(t)} - \eta \nabla_{\boldsymbol{a}} \psi(\boldsymbol{a}^{(t)}, \boldsymbol{b}^{(t)}) \right]$
3: $\quad \boldsymbol{b}^{(t+1/2)} = \Pi_{\mathcal{X}} \left[ \boldsymbol{b}^{(t)} + \eta \nabla_{\boldsymbol{b}} \psi(\boldsymbol{a}^{(t)}, \boldsymbol{b}^{(t)}) \right]$
4: $\quad \boldsymbol{a}^{(t+1)} = \Pi_{\mathcal{X}} \left[ \boldsymbol{a}^{(t)} - \eta \nabla_{\boldsymbol{a}} \psi(\boldsymbol{a}^{(t+1/2)}, \boldsymbol{b}^{(t+1/2)}) \right]$
5: $\quad \boldsymbol{b}^{(t+1)} = \Pi_{\mathcal{X}} \left[ \boldsymbol{b}^{(t)} + \eta \nabla_{\boldsymbol{b}} \psi(\boldsymbol{a}^{(t+1/2)}, \boldsymbol{b}^{(t+1/2)}) \right]$
6: **return** $(\boldsymbol{a}^{(t)}, \boldsymbol{b}^{(t)})_{t=0}^{T}$

---

convergence of extragradient descent methods in min-max optimization problems to obtain the following convergence guarantees for EDA in pseudo-games.[4]

**Theorem 3.4** (Convergence rate of EDA). *Consider a pseudo-game $\mathcal{G}$ with convex-concave cumulative regret that satisfies Assumption 3.3. Suppose that EDA (Algorithm 1) is run with $\eta < 1/\ell_{\nabla \psi}$, and that doing so generates the sequence of iterates $(\boldsymbol{a}^{(t)}, \boldsymbol{b}^{(t)})_{t=0}^{T}$. Let $\overline{\boldsymbol{a}^{(T)}} = 1/T \sum_{t=1}^{T} \boldsymbol{a}^{(t)}$ and $\overline{\boldsymbol{b}^{(T)}} = 1/T \sum_{t=1}^{T} \boldsymbol{b}^{(t)}$. Then the following convergence rate to a VE, i.e., to zero exploitability holds: $\max_{\boldsymbol{b} \in \mathcal{X}} \psi(\overline{\boldsymbol{a}^{(T)}}, \boldsymbol{b}) - \max_{\boldsymbol{a} \in \mathcal{X}} \psi(\boldsymbol{a}, \overline{\boldsymbol{b}^{(T)}}) \leq 1/T (d\ell_{\nabla \psi})$, where $d = \max_{(\boldsymbol{a}, \boldsymbol{b}) \in \mathcal{X} \times \mathcal{X}} \left\| (\boldsymbol{a}, \boldsymbol{b}) - (\boldsymbol{a}^{(0)}, \boldsymbol{b}^{(0)}) \right\|_{2}^{2}$. If, additionally, $\psi$ is $\mu$-strongly-convex-$\mu$-strongly-concave, and the learning rate $\eta = 1/4\ell_{\psi}$, then the following convergence bound also holds: $\max_{\boldsymbol{b} \in \mathcal{X}} \psi(\overline{\boldsymbol{a}^{(T)}}, \boldsymbol{b}) - \max_{\boldsymbol{a} \in \mathcal{X}} \psi(\boldsymbol{a}, \overline{\boldsymbol{b}^{(T)}}) \leq d\ell_{\psi}^{2}/\mu \left( 1 - \mu/4\ell_{\psi} \right)^{T}$.*

**Remark 3.5.** *If the players' utility functions have Lipschitz-continuous third-order derivatives, then convergence in last iterates can be obtained in $O(1/\varepsilon^{2})$ iterations [98].*

Since the complexity of two-player zero-sum convex-concave games, a special case of our model, is $\Omega(1/\varepsilon)$, the iteration complexity we derive is optimal. The exploitability minimization via EDA is thus an optimal approach to GNEP in convex-concave cumulative regret pseudo-games. Additionally, since pseudo-games generalize games, our results show that the complexity of this class of pseudo-games is the same as that of the corresponding games.

## 4   Pseudo-Games with Jointly Convex Constraints

More generally, the exploitability minimization problem is equivalent to a non-convex-concave min-max optimization problem, i.e., cumulative regret is non-convex-concave. In such cases, a minimax theorem does not hold, which precludes the existence of a saddle point, making the problem much harder. Solving non-convex-concave min-max optimization problems is NP-hard in general [99], so we instead set as our goal finding a (first-order) stationary point $\boldsymbol{a}^{*} \in \mathcal{X}$ of the exploitability $\varphi$, i.e., a point at which first-order deviations cannot decrease the exploitability, which we show can be found in polynomial-time. Note that any stationary point $\boldsymbol{a}^{*}$ of the exploitability is a $\varphi(\boldsymbol{a}^{*})$-GNE of the associated pseudo-game.

In recent years, a variety of methods have been developed that could be applied to finding stationary points of the exploitability [19, 25, 26, 27, 31]. Using the most efficient of these methods, one could obtain convergence to an $\varepsilon$-stationary point of the exploitability in $O(1/\varepsilon^{6})$ iterations [19]. However, as we will see, this rate would be very slow, and the convergence metric, not very strong, e.g., convergence in expected iterates. Instead, we demonstrate how to leverage the structure of the exploitability-minimization problem to obtain much faster convergence rates.

Unfortunately, the exploitability associated with the min-max characterization of pseudo-games with joint constraints is non-differentiable in general. This fact poses an obstacle when trying to design efficient algorithms that find its stationary points. However, by exploiting the structure of cumulative regret, we can regularize it to obtain a smooth objective. Observe the following: if

---

[4]All omitted proofs can be found in Appendix B.

$\boldsymbol{a}^* \in \arg\min_{\boldsymbol{a}\in\mathcal{X}} \max_{\boldsymbol{b}\in\mathcal{X}} \psi(\boldsymbol{a},\boldsymbol{b})$, then $\boldsymbol{a}^* \in \arg\max_{\boldsymbol{b}\in\mathcal{X}} \psi(\boldsymbol{a}^*,\boldsymbol{b})$. In other words, $\boldsymbol{b}^* = \boldsymbol{a}^*$ is a solution to the inner maximization problem. As a result, we can penalize exploitability in proportion to the distance between $\boldsymbol{a}$ and $\boldsymbol{b}$, while still ensuring that this penalized exploitability is minimized at a VE. We thus optimize the $\alpha$-**regularized cumulative regret** $\psi_\alpha : \mathcal{A} \times \mathcal{A} \to \mathbb{R}$, defined as $\psi_\alpha(\boldsymbol{a},\boldsymbol{b}) \doteq \psi(\boldsymbol{a},\boldsymbol{b}) - \frac{\alpha}{2}\|\boldsymbol{a}-\boldsymbol{b}\|_2^2$, whose associated $\alpha$-**regularized exploitability** $\varphi_\alpha : \mathcal{A} \to \mathbb{R}$ is given by $\varphi_\alpha(\boldsymbol{a}) = \max_{\boldsymbol{b}\in\mathcal{X}} \psi_\alpha(\boldsymbol{a},\boldsymbol{b})$. Von Heusinger and Kanzow show that a strategy profile $\boldsymbol{a}^*$ has no $\alpha$-regularized-exploitability, i.e., $\varphi_\alpha(\boldsymbol{a}^*) = 0$, iff $\boldsymbol{a}^*$ is a VE for all $\alpha > 0$ (Theorem 3.3 [65]).

To better understand the smoothness properties of the regularized exploitability, we present the **parametric Moreau envelope**, a Moreau envelope [100] in which the objective function is a function of the point w.r.t. which we regularize the objective function, which in our case is cumulative regret. Our next theorem is a generalization of the Moreau envelope theorem [100], which shows that $\alpha$-regularized exploitability is Lipschitz-smooth, by allowing us to see $\varphi_\alpha$ as a parametric Moreau envelope for $\psi$, an observation which to our knowledge has not been made in previous work. This Lipschitz-smoothness of $\varphi_\alpha$ gives rise to the possibility of deriving algorithms that converge to a stationary point of $\varphi_\alpha$, the key idea underlying our second algorithm (ADA; Algorithm 2).

**Theorem 4.1** (Parameteric Moreau Envelope Theorem)**.** *Given $\alpha > 0$, consider the parametric Moreau envelope $\varphi_\alpha(\boldsymbol{a}) \doteq \max_{\boldsymbol{b}\in\mathcal{X}} \left\{ \psi(\boldsymbol{a},\boldsymbol{b}) - \frac{\alpha}{2}\|\boldsymbol{a}-\boldsymbol{b}\|_2^2 \right\}$ and the associated proximal operator $\{\boldsymbol{b}^*(\boldsymbol{a})\} \doteq \arg\max_{\boldsymbol{b}\in\mathcal{X}} \left\{ \psi(\boldsymbol{a},\boldsymbol{b}) - \frac{\alpha}{2}\|\boldsymbol{a}-\boldsymbol{b}\|_2^2 \right\}$, where $\psi$ is $\ell_{\nabla\psi}$-Lipschitz smooth. Then $\varphi_\alpha(\boldsymbol{a})$ is $\left(\ell_{\nabla\psi} + \frac{\ell_{\nabla\psi}^2}{\alpha}\right)$-Lipschitz-smooth, with gradients $\nabla_{\boldsymbol{a}}\varphi_\alpha(\boldsymbol{a}) = \nabla_{\boldsymbol{a}}\psi(\boldsymbol{a},\boldsymbol{b}^*(\boldsymbol{a})) - \alpha(\boldsymbol{a}-\boldsymbol{b}^*(\boldsymbol{a}))$.*

Next, we formalize the definition of a stationary point. As the exploitability $\varphi_\alpha$ is in general non-convex, and the optimization problem $\min_{\boldsymbol{a}\in\mathcal{X}} \varphi_\alpha(\boldsymbol{a})$ is constrained, the first-order condition $\nabla\varphi_\alpha(\boldsymbol{a}) = 0$ is not sufficient for stationarity, since a solution on the boundary of the constraint set could also be first-order stationary. As a result, we use proximal mappings in our definition. In particular, we reformulate the constrained exploitability minimization problem as the unconstrained minimization problem $\min_{\boldsymbol{a}\in\mathbb{R}^{nm}} \varphi_\alpha(\boldsymbol{a}) + \mathbb{1}_{\mathcal{X}}(\boldsymbol{a})$. In this optimization problem, $\varphi_\alpha$ is continuous and Lipschitz smooth, while $\mathbb{1}_{\mathcal{X}}$ is continuous and convex. Additionally, the set of solutions is non-empty. As a result, one can solve this problem using the proximal gradient method, which, in this case, is equivalent to the projected gradient method [97, 101]. This view of the problem as proximal minimization allows us to provide a clear definition of a first-order stationary point.

Define the **projected gradient operator** of a function $\varphi : \mathcal{A} \to \mathbb{R}$ for some step size $\eta_{\boldsymbol{a}} > 0$ as follows: $G^\varphi_{\eta_{\boldsymbol{a}}}(\boldsymbol{a}) = \boldsymbol{a} - \Pi_{\mathcal{X}}\left[\boldsymbol{a} - \eta_{\boldsymbol{a}}\nabla\varphi(\boldsymbol{a})\right]$. The projected gradient operator is a special case of the gradient mapping operator, and a generalization of the gradient that is projected back onto the feasible set $\mathcal{X}$ [101]. The zeros of the projected gradient operator capture stationary points both at the boundary of the set $\mathcal{X}$ and in its interior. Indeed, $\boldsymbol{a} \in \mathcal{A}$ is a **stationary point** of $\varphi$ if $G^\varphi_{\eta_{\boldsymbol{a}}}(\boldsymbol{a}) = \boldsymbol{0}$. Note that, for an action profile $\boldsymbol{a}^* \in \mathcal{A}$ to be a VE, it is necessary that it is stationary with respect to the VE exploitability (Proposition 3, [80]). Our goal will thus be to achieve convergence to a strategy profile $\boldsymbol{a} \in \mathcal{X}$ s.t. $G^{\varphi_\alpha}_{\eta_{\boldsymbol{a}}}(\boldsymbol{a}) = 0$.

ADA (Algorithm 2) is a nested GDA algorithm, similar to those of Nouiehed et al. [25] and Goktas and Greenwald [85], with the objective function augmented by a smart regularizer. The algorithm is reminiscent of EDA, interleaving gradient ascent and descent steps. The key difference is that whereas EDA runs only a single step of extragradient ascent (to find a better response), ADA runs multiple steps of gradient ascent to approximate a best response. Indeed, as Theorem 4.2 shows, ADA's accuracy depends on the accuracy of the best-response found.

To derive convergence rates for ADA, we rely on two technical lemmas. The first tells us that the inner loop can approximate the gradient of the exploitability to any desired precision. More specifically, we bound the error $\left\|\nabla_{\boldsymbol{a}}\varphi_\alpha(\boldsymbol{a}^{(t)}) - \nabla_{\boldsymbol{a}}\psi_\alpha(\boldsymbol{a}^{(t)},\boldsymbol{b}^{(t)})\right\| \leq \varepsilon$ by providing a lower bound on the number of inner loop iterations $T_{\boldsymbol{b}}$ necessary to achieve a desired precision $\varepsilon$, for all outer loop iterations $t \in [T_{\boldsymbol{a}}]$ (Lemma B.3). The second lemma bounds the algorithm's progress at each iteration of the outer loop by relating $\varepsilon$ and $G^{\varphi_\alpha}_{\eta_{\boldsymbol{a}}}(\boldsymbol{a}_t)$ (Lemma B.4). Finally, by telescoping the inequalities in the progress lemma, we obtain our main theorem: The best iterate found by

---
**Algorithm 2** Augmented Descent Ascent (ADA)
---
**Inputs:** $n, \boldsymbol{u}, \boldsymbol{g}, \alpha, \eta_{\boldsymbol{a}}, \eta_{\boldsymbol{b}}, T_{\boldsymbol{a}}, T_{\boldsymbol{b}}, \boldsymbol{a}^{(0)}, \boldsymbol{b}^{(0)}$

**Outputs:** $(\boldsymbol{a}^{(t)}, \boldsymbol{b}^{(t)})_{t=0}^{T_{\boldsymbol{a}}}$

  1: **for** $t = 0, \ldots, T_{\boldsymbol{a}} - 1$ **do**

  2:      $\boldsymbol{a}^{(t+1)} = \Pi_{\mathcal{X}} \left[ \boldsymbol{a}^{(t)} - \eta_{\boldsymbol{a}} \left( \nabla_{\boldsymbol{a}} \psi(\boldsymbol{a}^{(t)}, \boldsymbol{b}^{(t)}) - \alpha(\boldsymbol{a}^{(t)} - \boldsymbol{b}^{(t)}) \right) \right]$

  3:      $\boldsymbol{b} = \boldsymbol{0}$

  4:      **for** $s = 0, \ldots, T_{\boldsymbol{b}} - 1$ **do**

  5:          $\boldsymbol{b} = \Pi_{\mathcal{X}} \left[ \boldsymbol{b} + \eta_{\boldsymbol{b}} \left( \nabla_{\boldsymbol{b}} \psi(\boldsymbol{a}^{(t+1)}, \boldsymbol{b}) - \alpha(\boldsymbol{a}^{(t+1)} - \boldsymbol{b}) \right) \right]$

  6:      $\boldsymbol{b}^{(t+1)} = \boldsymbol{b}$

  7: **return** $(\boldsymbol{a}^{(t)}, \boldsymbol{b}^{(t)})_{t=0}^{T_{\boldsymbol{a}}}$

---

ADA, as measured by the norm of the projected gradient operator, converges to an approximate stationary point of the exploitability in all Lipschitz-smooth pseudo-games with jointly convex constraints. We note that if the gradient of the regularized exploitability can be computed exactly, then convergence to an exact stationary point can be obtained.

**Theorem 4.2** (Convergence to Stationary Point of Exploitability). *Suppose that ADA is run on a pseudo-game $\mathcal{G}$ which satisfies Assumption 3.3 with learning rates $\eta_{\boldsymbol{a}} > \frac{2}{\ell_{\nabla \psi_\alpha} + \frac{\ell_{\nabla \psi_\alpha}^2}{\alpha}}$ and $\eta_{\boldsymbol{b}} = \frac{1}{\ell_{\psi_\alpha}}$,*

*for any number of outer loop iterations $T_{\boldsymbol{a}} \in \mathbb{N}_{++}$ and for $T_{\boldsymbol{b}} \geq \frac{2 \log\left( \frac{\varepsilon}{\ell_{\nabla \psi_\alpha}} \sqrt{\frac{2\alpha}{c}} \right)}{\log\left( \frac{\alpha}{\ell_{\nabla \psi_\alpha}} \right)}$ total inner loop*

*iterations where $\varepsilon > 0$. Then the outputs $(\boldsymbol{a}^{(t)}, \boldsymbol{b}^{(t)})_{t=0}^{T}$ satisfy $\min_{t=0,\ldots,T_{\boldsymbol{a}}-1} \left\| G_{\eta_{\boldsymbol{a}}}^{\varphi_\alpha}(\boldsymbol{a}^{(t)}) \right\|_2^2 \leq$*

$$
\frac{1}{\frac{1}{\eta_{\boldsymbol{a}}} - \frac{\ell_{\nabla \psi_\alpha} + \frac{\ell_{\nabla \psi_\alpha}^2}{\alpha}}{2}} \left( \frac{\varphi_\alpha(\boldsymbol{a}^{(0)})}{T_{\boldsymbol{a}}} + \varepsilon \left( \left( \ell_{\nabla \psi_\alpha} + \frac{\ell_{\nabla \psi_\alpha}^2}{\alpha} + \varepsilon \right) \left( \frac{\ell_{\nabla \psi_\alpha} + \frac{\ell_{\nabla \psi_\alpha}^2}{\alpha}}{2} - \frac{1}{\eta_{\boldsymbol{a}}} \right) + \ell_{\psi_\alpha} \right) \right) \ .
$$

Note that the stationary points to which ADA converges are global minima of $\varphi_\alpha$ iff $\varphi_\alpha$ is invex, in which case ADA converges to a VE, and hence a GNE [102]. We can also obtain faster and last iterate convergence to VE in a more restricted class of games. A pseudo-game is said to be a $\mu$-**PL-pseudo-game** if the regularized exploitability $\varphi_\alpha$ satisfies the $\mu$-**projection-Polyak-Lojasiewicz (PL)** condition, i.e., the projected gradient operator associated with $\varphi_\alpha$ satisfies $^{1}/_{2} \left\| G_{\eta_{\boldsymbol{a}}}^{\varphi_\alpha}(\boldsymbol{a}) \right\|_2^2 \geq \mu \left( \varphi_\alpha(\boldsymbol{a}) - \min_{\boldsymbol{a} \in \mathcal{X}} \varphi_\alpha(\boldsymbol{a}) \right)$ for all $\boldsymbol{a} \in \mathcal{X}$.[5] A similar PL-condition for games was recently used by Raghunathan, Cherian, and Jha [104], which they argued is natural.

**Theorem 4.3** (PL Exploitability Convergence). *Suppose ADA is run on a PL-pseudo-game $\mathcal{G}$ which satisfies Assumption 3.3 with learning rates $\eta_{\boldsymbol{a}} \in \left[ \ell_{\nabla \psi_\alpha} + \frac{\ell_{\nabla \psi_\alpha}^2}{\alpha}, \ell_{\nabla \psi_\alpha} + \frac{\ell_{\nabla \psi_\alpha}^2}{\alpha} + \frac{1}{2\mu} \right]$ and $\eta_{\boldsymbol{b}} = {}^{1}/_{\ell_{\nabla \psi_\alpha}}$, for any number of outer loop iterations $T_{\boldsymbol{a}} \in \mathbb{N}_{++}$ and for $T_{\boldsymbol{b}} \geq \frac{\log\left( \frac{\varepsilon}{\nabla \psi_\alpha} \sqrt{\frac{2\alpha}{c}} \right)}{\log\left( \frac{\alpha}{\nabla \psi_\alpha} \right)}$ total inner loop iterations. Then the outputs $(\boldsymbol{a}^{(t)}, \boldsymbol{b}^{(t)})_{t=0}^{T}$ satisfy $\varphi_\alpha(\boldsymbol{a}^{(T_{\boldsymbol{a}})}) \leq \left[ 1 + 2\mu \left( \frac{\ell_{\nabla \psi_\alpha} + \frac{\ell_{\nabla \psi_\alpha}^2}{\alpha}}{2} - \frac{1}{\eta_{\boldsymbol{a}}} \right) \right]^{T_{\boldsymbol{a}}} \varphi_\alpha(\boldsymbol{a}^{(0)}) +$*

$$
\left( \left( \ell_{\nabla \psi_\alpha} + \frac{\ell_{\nabla \psi_\alpha}^2}{\alpha} + \varepsilon \right) \left( \frac{\ell_{\nabla \psi_\alpha} + \frac{\ell_{\nabla \psi_\alpha}^2}{\alpha}}{2} - \frac{1}{\eta_{\boldsymbol{a}}} \right) + \ell_{\psi_\alpha} \right) \varepsilon.
$$

We note that this result also implies convergence to a VE in strongly-convex exploitability games in linear time as well, since strong convexity implies the PL-condition.

---

[5]We note that this is a special case of the proximal-Polyak-Lojasiewicz condition (See Section 4, [103]).

## 5 Experiments

In this section, we report on experiments that demonstrate the effectiveness of our algorithms as compared to others that were also designed to compute GNE. We present numerical results on benchmark psuedo-games that were studied empirically in previous work [104]: 1) monotone pseudo-games with affine constraints and 2) bilinear pseudo-games with jointly convex constraints.[6] We compare our algorithms to the **accelerated mirror-prox quadratic penalty method** (AMPQP), the only GNE-finding algorithm with theoretical convergence guarantees *and rates* [78].[7] For the AMPQP algorithm, we use the hyperparameter settings derived in theory when available, and otherwise conduct a grid search for the best ones. We compare the convergence rates of our algorithms to that of AMPQP, which converges at the same rate as EDA, but slower than ADA, in settings in which it is guaranteed to converge, i.e., in monotone pseudo-games.

In the monotone pseudo-games, for EDA we use a constant learning rate of $\eta = 0.02$, while for ADA we use the constant learning rates of $\eta_{\boldsymbol{a}} = 0.02$ and $\eta_{\boldsymbol{a}} = 0.05$. These rates approximate the Lipschitz-smoothness parameter of the pseudo-games. In the bilinear pseudo-games, we use the theoretically-grounded (Theorem 3.4) learning rate of $\eta = 1/\|\boldsymbol{Q}\|$ for EDA, and $\eta_{\boldsymbol{a}} = \|\boldsymbol{Q}\| + \frac{\|\boldsymbol{Q}\|^2}{\alpha}$ and $\eta_{\boldsymbol{b}} = 1/\|\boldsymbol{Q}\|$ for ADA. In both settings, we use a regularization of $\alpha = 0.1$ for ADA, which we found by grid search. Likewise, in our implementation of AMPQP, grid search led us to initialize $\beta_0 = 0.01, \gamma = 1.05, \alpha = 0.2$ in both settings.[8] Additionally, because the iterates generated by AMPQP are not necessarily feasible, we projected them onto the feasible set of actions, so as to avoid infinite exploitability. This heuristic improved the convergence rate of AMPQP significantly. The first iterate for all methods was common, and initialized at random.

**Monotone Games with Jointly Affine Constraints**    We begin by experimenting with monotone games with jointly affine constraints, the largest class of pseudo-games for which convergence rates exist for AMPQP. In particular, we consider norm-minimization pseudo-games with payoff functions $u_i(\boldsymbol{a}) = -\left\|\sum_{i \in [n]} \boldsymbol{a}_i - \boldsymbol{s}_i\right\|$, for all players $i \in [n]$, and for some randomly initialized shifting factors $\boldsymbol{s}_i$, as well as jointly affine constraints $g(\boldsymbol{a}) = 1 - \sum_{j \in [m]} a_{1j} - \sum_{i \in [m]} a_{2j}$, with action spaces $\mathcal{A}_i = [-10, 10]^m$, for all players $i \in [n]$. Although norm-minimization games are trivial, in the sense that they can be solved in closed form, this is not the case for pseudo-games; on the contrary, they form a bedrock example for monotone pseudo-games ([57], Example 1).

In Figure 1a, we observe that all three algorithms converge to a GNE. Interestingly, EDA and ADA find distinct GNEs from AMPQP, which is not entirely surprising, as AMPQP is not an exploitability minimization algorithm. Although all the algorithms eventually find a GNE, exploitability does not decrease monotonically (Figure 1b); on the contrary, it increases before eventually decreasing to zero. Convergence to a GNE is expected for AMPQP, as per the theory. EDA, however, is not guaranteed to converge to a GNE in pseudo-games beyond convex exploitability; yet, we still observe convergence, albeit at a slower rate than AMPQP or ADA. Likewise, ADA converges to a GNE, although it is again not guaranteed, much faster than AMPQP.

**Bilinear General-Sum Games with Jointly Convex Constraints**    Second, we consider the class of two-player general-sum games with jointly convex constraints with payoff functions $u_1(\boldsymbol{a}) = \boldsymbol{a}_1^T \boldsymbol{Q}_1 \boldsymbol{a}_2, u_2(\boldsymbol{a}) = \boldsymbol{a}_1^T \boldsymbol{Q}_2 \boldsymbol{a}_2, g(\boldsymbol{a}) = 1 - \|\boldsymbol{a}\|_2^2, \mathcal{A}_1 = \mathcal{A}_2 = [-10, 10]^m$. Such games are not monotone, and hence AMPQP is not guaranteed to converge in theory. In Figure 1c, we see that convergence is not guaranteed empirically either; shown is a sample run in which ADA and EDA converge, while AMPQP exhibits erratic behavior. Non-convergence for AMPQP was

---

[6]We describe additional experiments in the supplemental work, with two-player zero-sum bilinear pseudo-games with jointly affine constraints and two-player zero-sum bilinear pseudo-games with jointly convex constraints. The results are not qualitatively different than those presented here. All our code can be found on https://github.com/denizalp/exploit-min.

[7]We note that we are not reporting on our results with the Accelerated Mirror-Prox Augmented Lagrangian Method (AMPAL), another method studied in the literature with the convergence rates as AMPQP, because we found this algorithm to be unstable on our benchmark pseudo-games.

[8]As suggested by Jordan, Lin, and Zampetakis [78], we replaced the convergence tolerance parameter $\delta$, with the number of iterations for which the accelerated mirror-prox subroutine [105] was run. Then, at each iteration of AMPQP, this parameter was increased by a factor of $\gamma$ of AMPQP.

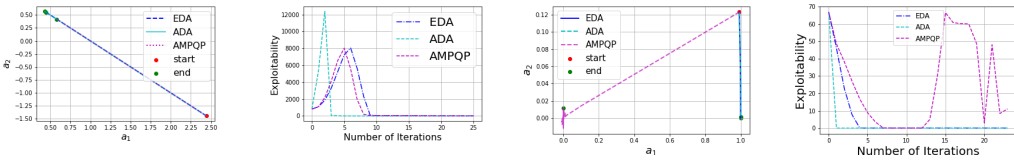

(a) Phase portrait for a 2 player monotone game with $m = 1$

(b) Exploitability for a 5 player monotone game with $m = 10$

(c) Phase portrait for bilinear general-sum pseudo-game with $m = 1$

(d) Exploitability for a bilinear general-sum pseudo-game; $m = 10$

Figure 1: Convergence in monotone and bilinear general-sum pseudo-games.

observed in 93 of the 100 experiments run. In fact, even though AMPQP gets very close a GNE (Figure 1d), i.e., zero exploitability, it does not stabilize there. We likewise found that ADA and EDA do not converge to a GNE from all random initializations. In such cases, we restarted the algorithms with a new random initialization; this process lead to on average 11 restarts for EDA and 10 restarts for ADA, meaning both algorithms converged to a GNE within this window.

## 6  Conclusion

In this paper, we provide an affirmative answer to an open question first posed by Flam and Ruszczynski [80], who suggested that projected gradient methods could be developed to minimize exploitability in pseudo-games. Although some existing methods [57] make use of the exploitability concept to analyze algorithms that take alternative optimization approaches, we know of none that are exploitability-minimization algorithms per se, as exploitability itself is not explicitly minimized. Our main contribution is the observation that the exploitability minimization problem can be recast as a simple and seemingly trivial min-max optimization problem (Observation 3.2). Our analysis effectively extends the class of games which can be solved as potential games, since the exploitability can be seen as a potential for any game.

We provide a definition of stationarity for exploitability via the gradient mapping operator from proximal theory, and we use this definition to characterize the limit points of our algorithms. Our characterization relies on novel proof techniques inspired by recent analyses of algorithms for min-max optimization problems [25], as well as a parameterized version of Moreau's envelope theorem, which may be of independent interest. This approach, which takes advantage of the structural properties of exploitability, allows us to rigorously analyze an algorithm that converges to stationary points of the VE exploitability in all Lipschitz-smooth pseudo-games with jointly convex constraints, and obtain algorithms which are orders of magnitude faster than known min-max optimization algorithms for similar types of problems. To summarize, our results 1. extend known polynomial-time convergence guarantees to VEs, and thus GNEs, to a class of pseudo-games beyond monotone, and 2. provide a general approach to approximate, with convergence guarantees, solutions to Lipschitz-smooth pseudo-games with jointly convex constraints.

The design of efficient algorithms for constrained non-convex-concave min-max optimization problems is an open problem. The literature thus far has focused on problems in which only the variable to be maximized is constrained [25, 27, 29]. Our techniques may provide a path to solving non-convex-concave min-max optimization problems in entirely constrained domains, since our algorithms solve a non-convex-concave optimization problem where both variables are constrained.

## Acknowledgments and Disclosure of Funding

The ideas in this paper stemmed from conversations and seminars held during the Learning in Games workshop at the Simons Institute for the Theory of Computing. This work was also supported by NSF Grant CMMI-1761546.

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
