# A   Additional Experiments and Detailed Setup

Our experimental goals were to understand the performance of our algorithms relative to the state of the art methods in practice. We present numerical results on benchmark psuedo-games that were studied empirically in previous work [104]: 1) two-player zero-sum bilinear pseudo-games with jointly affine constraints; 2) two-player zero-sum bilinear pseudo-games with jointly convex constraints; 3) monotone games with affine constraints, and 4) bilinear games with jointly convex constraints. The first two set of experiments can be found in this section, while the two other experiments can be found in Section 5.

We compare our algorithms to the **accelerated mirror-prox quadratic penalty method** (AMPQP, Algorithm 1 [78]), the only GNE-finding algorithm with theoretical convergence guarantees *and rates* [78].[9] For the AMPQP algorithm, we use the hyperparameter settings derived in theory when available, and otherwise conduct a grid search for the best ones. We compare the convergence rates of our algorithms to that of AMPQP, which converges at the same rate as EDA, but slower than ADA, in settings in which it is guaranteed to converge, i.e., in monotone pseudo-games.

**Computational Resources**   Our experiments were run on a MacOS machine with 8GB RAM and an Apple M1 chip, and took about 1 hour to run. Only CPU resources were used.

**Programming Languages, Packages, and Licensing**   We ran our experiments in Python 3.7 [106], using NumPy [107], and CVXPY [108]. All figures were graphed using Matplotlib [109].

Numpy is distributed under a liberal BSD license. Matplotlib only uses BSD compatible code, and its license is based on the PSF license. CVXPY is licensed under an APACHE license.

**Common Experimental Details**   All our algorithms were run to generate 50 iterates in total. In particular, we had $T = 50$ for EDA, $T_{\boldsymbol{a}} = T_{\boldsymbol{b}} = 50$. For AMPQP, we ran the main routine (Algorithm 1, [78]) for 50 iterations, i.e., $T = 50$. As the iterates generated by AMPQP are not necessarily feasible, we projected them onto the feasible set of actions, so as to avoid unbounded exploitability. This heuristic improved the convergence rate of AMPQP significantly. The first iterate for all methods was common, and initialized at random.

**Code repository**   The data our experiments generated, as well as the code used to produce our visualizations, can be found in our code repository.

In what follows, we present two additional set of experiments comparing the empirical convergence rates of all three algorithms in zero-sum games.

**Bilinear Two-Player Zero-Sum Game with Jointly Affine Constraints**   We consider the following two player pseudo-game with $m \in \mathbb{N}_+$ pure strategies: $u_1(\boldsymbol{a}) = \boldsymbol{a}_1{}^T \boldsymbol{Q} \boldsymbol{a}_2 = -u_2(\boldsymbol{a}), g(\boldsymbol{a}) = 1 - \sum_{j \in [m]} a_{1j} - \sum_{i \in [m]} a_{2j}, \mathcal{A}_1 = \mathcal{A}_2 = [-10, 10]^m$, where $u_1, u_2$ are payoff functions of the players, $g$ is the joint constraint function for all players, and $\mathcal{A}_1, \mathcal{A}_2$, are the players' strategy spaces. A game version of this pseudo-game was explored by Gidel et al. [110] and Raghunathan, Cherian, and Jha [104]. Current convergence guarantees for GNE-finding algorithms are known only for the type of jointly affine constraints $g$ that we consider in this example [78].

For EDA, we use a constant learning rate of $\eta = 1/\|\boldsymbol{Q}\|$ as determined by our theory (Theorem 3.4). Similarly, for ADA, we use the theoretically-grounded constant learning rates of $\eta_{\boldsymbol{a}} = 1/\|\boldsymbol{Q}\| + \frac{\|\boldsymbol{Q}\|^2}{\alpha}$ and $\eta_{\boldsymbol{a}} = 1/\|\boldsymbol{Q}\|$, together with a regularization of $\alpha = 0.1$, which was determined by grid search. In our implementation of AMPQP, grid search led us to initialize $\beta_0 = 0.01, \gamma = 1.05, \alpha = 0.2$.[10]

In these experiments, the initialization seemed to have little impact on the algorithms' performances.

---

[9]We note that we are not reporting on our results with the Accelerated Mirror-Prox Augmented Lagrangian Method (AMPAL), another method studied in the literature with the convergence rates as AMPQP, because we found this algorithm to be unstable on our benchmark pseudo-games.

[10]As suggested by Jordan, Lin, and Zampetakis [78], we replaced the convergence tolerance parameter $\delta$ in Algorithm 1 of Jordan, Lin, and Zampetakis [78], with a number of iterations for which the accelerated mirror-prox subroutine algorithm [105] was run. which at eachFor every iteration of AMPQP, this parameter was increased by a factor of $\gamma$ of AMPQP.

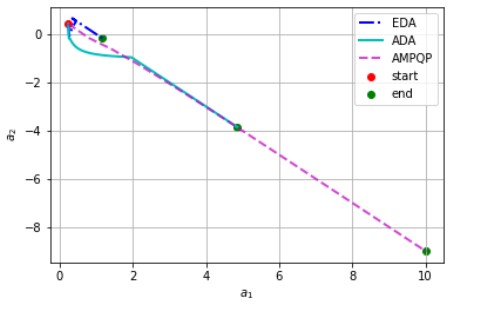
(a) Sample phase portrait for $m = 1$

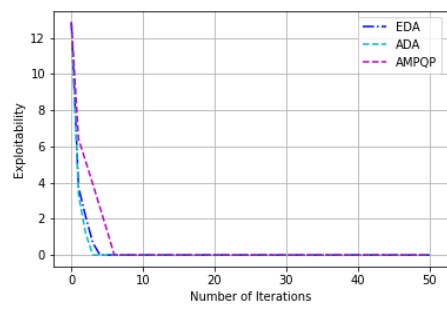
(b) Sample exploitability convergence for $m = 10$

Figure 2: Convergence in zero-sum bilinear pseudo-game with jointly affine constraints.

In Figure 2, we plot the phase portrait of a randomly initialized one-dimensional pseudo-game, as well as the convergence rates in a higher-dimensional pseudo-game. We see that all methods converge sublinearly, with EDA and AMPQP at similar rates, and ADA slightly faster. Interestingly, the three methods find distinct GNEs (Figure 2a). This outcome can be explained by the fact that the three algorithms have different objectives. In the next set of experiments we will see that although AMPQP converges to a different GNE than that of EDA and ADA when the constraints are not affine, EDA and ADA (which have similar objectives) converge to the same GNE.

**Bilinear Two-Player Zero-Sum Game with Jointly Convex Constraints** Next, we consider the same setting as above, replacing the affine constraints with convex $l_2$-ball constraints, i.e., $g(\boldsymbol{a}) = 1 - \|\boldsymbol{a}\|_2^2$. This setting is of interest since AMPQP is not guaranteed to converge in pseudo-games with non-affine constraints. Nonetheless, we observe that AMPQP converges, although this requires a change in the parameters we use, namely we chose $\beta_0 = 0.01, \gamma = 1.05, \alpha = 0.2$ using grid search. Additionally, we observed that changing the EDA regularization parameter to $\alpha = 1$ resulted in faster convergence. This suggests that the convexity of the constraints affects the structure of the regularized exploitability, and points to a need for additional theory.

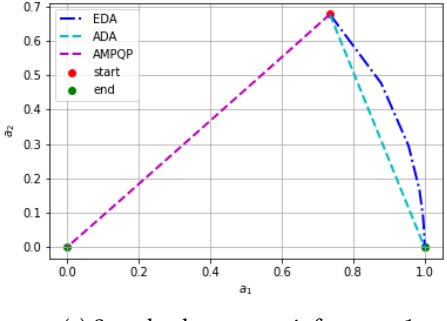
(a) Sample phase portrait for $m = 1$

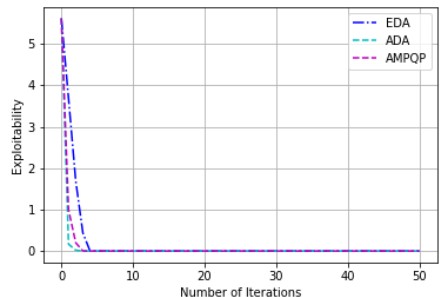
(b) Sample exploitability convergence for $m = 10$

Figure 3: Convergence in zero-sum bilinear pseudo-game with jointly convex constraints.

Once again, we observe in Figure 2b that all three algorithms converge at a sublinear rate, with AMPQP and EDA at a similar rate, and ADA slightly faster. Interestingly, in Figure 3a, the GNEs found by ADA and EDA are the same in these experiments, although ADA seems to find a more direct path to there. This could be a result of the regularization, which penalizes large deviations from the players' best-responses at each iteration of the algorithm.

# B Omitted Proofs

**Preliminaries** A **min-max Stackelberg game**, denoted $(\mathcal{X}, \mathcal{Y}, f, \boldsymbol{g})$, is a two-player, zero-sum game, where one player, who we call the $\boldsymbol{x}$-player (resp. the $\boldsymbol{y}$-player), is trying to minimize their loss (resp. maximize their gain), defined by a continuous **objective function** $f : X \times Y \to \mathbb{R}$, by choosing a strategy from a compact **strategy set** $\mathcal{X} \subset \mathbb{R}^n$ (resp. $\mathcal{Y} \subset \mathbb{R}^m$) s.t. $\boldsymbol{g}(\boldsymbol{x}, \boldsymbol{y}) \geq 0$ where $\boldsymbol{g}(\boldsymbol{x}, \boldsymbol{y}) = (g_1(\boldsymbol{x}, \boldsymbol{y}), \dots, g_d(\boldsymbol{x}, \boldsymbol{y}))^T$ with $g_k : \mathcal{X} \times \mathcal{Y} \to \mathbb{R}$, for all $k \in [d]$. A strategy profile $(\boldsymbol{x}, \boldsymbol{y}) \in \mathcal{X} \times \mathcal{Y}$ is said to be **feasible** iff $g_k(\boldsymbol{x}, \boldsymbol{y}) \geq 0$, for all $k \in [d]$. The function $f$ maps a pair of feasible strategies taken by the players to a real value (i.e., a payoff), which represents the loss (resp. the gain) of the $\boldsymbol{x}$-player (resp. $\boldsymbol{y}$-player). The relevant solution concept for Stackelberg games is the **Stackelberg equilibrium (SE)**. A strategy profile $(\boldsymbol{x}^*, \boldsymbol{y}^*) \in \mathcal{X} \times \mathcal{Y}$ s.t. $\boldsymbol{g}(\boldsymbol{x}^*, \boldsymbol{y}^*) \geq \boldsymbol{0}$ is a **Stackelberg equilibrium** if $\max_{\boldsymbol{y} \in \mathcal{Y}: \boldsymbol{g}(\boldsymbol{x}^*, \boldsymbol{y}) \geq 0} f(\boldsymbol{x}^*, \boldsymbol{y}) \leq f(\boldsymbol{x}^*, \boldsymbol{y}^*) \leq \min_{\boldsymbol{x} \in \mathcal{X}} \max_{\boldsymbol{y} \in \mathcal{Y}: \boldsymbol{g}(\boldsymbol{x}, \boldsymbol{y}) \geq 0} f(\boldsymbol{x}, \boldsymbol{y})$.

**Theorem 3.4** (Convergence rate of EDA). *Consider a pseudo-game $\mathcal{G}$ with convex-concave cumulative regret that satisfies Assumption 3.3. Suppose that EDA (Algorithm 1) is run with $\eta < 1/\ell_{\nabla \psi}$, and that doing so generates the sequence of iterates $(\boldsymbol{a}^{(t)}, \boldsymbol{b}^{(t)})_{t=0}^T$. Let $\overline{\boldsymbol{a}^{(T)}} = 1/T \sum_{t=1}^T \boldsymbol{a}^{(t)}$ and $\overline{\boldsymbol{b}^{(T)}} = 1/T \sum_{t=1}^T \boldsymbol{b}^{(t)}$. Then the following convergence rate to a VE, i.e., to zero exploitability holds: $\max_{\boldsymbol{b} \in \mathcal{X}} \psi(\overline{\boldsymbol{a}^{(T)}}, \boldsymbol{b}) - \max_{\boldsymbol{a} \in \mathcal{X}} \psi(\boldsymbol{a}, \overline{\boldsymbol{b}^{(T)}}) \leq 1/T (d\ell_{\nabla \psi})$, where $d = \max_{(\boldsymbol{a}, \boldsymbol{b}) \in \mathcal{X} \times \mathcal{X}} \|(\boldsymbol{a}, \boldsymbol{b}) - (\boldsymbol{a}^{(0)}, \boldsymbol{b}^{(0)})\|_2^2$. If, additionally, $\psi$ is $\mu$-strongly-convex-$\mu$-strongly-concave, and the learning rate $\eta = 1/4\ell_\psi$, then the following convergence bound also holds: $\max_{\boldsymbol{b} \in \mathcal{X}} \psi(\overline{\boldsymbol{a}^{(T)}}, \boldsymbol{b}) - \max_{\boldsymbol{a} \in \mathcal{X}} \psi(\boldsymbol{a}, \overline{\boldsymbol{b}^{(T)}}) \leq d\ell_\psi^2/\mu \left(1 - \mu/4\ell_\psi\right)^T$.*

*Proof.* The results follow from the results of Nemirovski [21]. $\qquad \square$

**Lemma 3.1.** *Given a pseudo-game $\mathcal{G}$, for all $\boldsymbol{a} \in \mathcal{A}$, $\varphi(\boldsymbol{a}) \geq 0$. Additionally, a strategy profile $\boldsymbol{a}^* \in \mathcal{X}(\boldsymbol{a}^*)$ is a GNE iff it achieves the lowerbound, i.e., $\varphi(\boldsymbol{a}^*) = 0$.*

*Proof.* We first prove the second part of the lemma.

(GNE $\implies$ No-Exploitability): Suppose that $\boldsymbol{a}^* \in \mathcal{X}(\boldsymbol{a}^*)$ is a GNE, i.e., for all players $i \in [n]$, $\boldsymbol{a}_i^* \in \arg\max_{\boldsymbol{b}_i \in \mathcal{X}_i(\boldsymbol{a}_{-i}^*)} u_i(\boldsymbol{b}_i, \boldsymbol{a}_{-i}^*)$. Then, for all players $i \in [n]$, we have:

$$\max_{\boldsymbol{b}_i \in \mathcal{X}_i(\boldsymbol{a}_{-i}^*)} u_i(\boldsymbol{b}_i, \boldsymbol{a}_{-i}^*) - u_i(\boldsymbol{a}^*) = 0 \tag{1}$$

Summing up across all players $i \in [n]$, we get:

$$\sum_{i \in [n]} \left( \max_{\boldsymbol{b}_i \in \mathcal{X}_i(\boldsymbol{a}_{-i}^*)} u_i(\boldsymbol{b}_i, \boldsymbol{a}_{-i}^*) - u_i(\boldsymbol{a}^*) \right) = 0 \tag{2}$$

$$\sum_{i \in [n]} \max_{\boldsymbol{b}_i \in \mathcal{X}_i(\boldsymbol{a}_{-i}^*)} \left( u_i(\boldsymbol{b}_i, \boldsymbol{a}_{-i}^*) - u_i(\boldsymbol{a}_i^*, \boldsymbol{a}_{-i}^*) \right) = 0 \tag{3}$$

$$\sum_{i \in [n]} \max_{\boldsymbol{b}_i \in \mathcal{X}_i(\boldsymbol{a}_{-i}^*)} \text{Regret}_i(\boldsymbol{a}_i^*, \boldsymbol{b}_i; \boldsymbol{a}_{-i}^*) = 0 \tag{4}$$

$$\max_{\boldsymbol{b} \in \mathcal{X}(\boldsymbol{a}^*)} \sum_{i \in [n]} \text{Regret}_i(\boldsymbol{a}_i^*, \boldsymbol{b}_i; \boldsymbol{a}_{-i}^*) = 0 \tag{5}$$

$$\max_{\boldsymbol{b} \in \mathcal{X}(\boldsymbol{a}^*)} \psi(\boldsymbol{a}^*, \boldsymbol{b}) = 0 \tag{6}$$

(No-Exploitability $\implies$ GNE):

Suppose that we have $\boldsymbol{a}^* \in \mathcal{X}(\boldsymbol{a}^*)$ such that $\varphi(\boldsymbol{a}^*) = 0$, then:

$$\varphi(\boldsymbol{a}^*) = 0 \tag{7}$$

$$\max_{\boldsymbol{b} \in \mathcal{X}(\boldsymbol{a}^*)} \sum_{i \in [n]} \operatorname{Regret}_i(\boldsymbol{a}_i^*, \boldsymbol{b}_i; \boldsymbol{a}_{-i}^*) = 0 \tag{8}$$

$$\sum_{i \in [n]} \max_{\boldsymbol{b}_i \in \mathcal{X}_i(\boldsymbol{a}_{-i}^*)} \operatorname{Regret}_i(\boldsymbol{a}_i^*, \boldsymbol{b}_i; \boldsymbol{a}_{-i}^*) = 0 \tag{9}$$

$$\sum_{i \in [n]} \max_{\boldsymbol{b}_i \in \mathcal{X}_i(\boldsymbol{a}_{-i}^*)} \left( u_i(\boldsymbol{b}_i, \boldsymbol{a}_{-i}^*) - u_i(\boldsymbol{a}_i^*, \boldsymbol{a}_{-i}^*) \right) = 0 \tag{10}$$

$$\sum_{i \in [n]} \left( \max_{\boldsymbol{b}_i \in \mathcal{X}_i(\boldsymbol{a}_{-i}^*)} u_i(\boldsymbol{b}_i, \boldsymbol{a}_{-i}^*) - u_i(\boldsymbol{a}^*) \right) = 0 \tag{11}$$

We remark that since $\boldsymbol{a}_i^* \in \mathcal{X}_i(\boldsymbol{a}_{-i})$, we have for all $i \in [n]$, $\max_{\boldsymbol{b}_i \in \mathcal{X}_i(\boldsymbol{a}_{-i}^*)} u_i(\boldsymbol{b}_i, \boldsymbol{a}_{-i}^*) - u_i(\boldsymbol{a}^*) \geq u_i(\boldsymbol{a}^*) - u_i(\boldsymbol{a}^*) \geq 0$. As a result, we must have that for all players $i \in [n]$:

$$u_i(\boldsymbol{a}^*) = \max_{\boldsymbol{b}_i \in \mathcal{X}_i(\boldsymbol{a}_{-i}^*)} u_i(\boldsymbol{b}_i, \boldsymbol{a}_{-i}^*) \tag{12}$$

The first part of the lemma follows from the previous remark. $\qquad\square$

**Theorem 4.1** (Parameteric Moreau Envelope Theorem). *Given $\alpha > 0$, consider the parametric Moreau envelope $\varphi_\alpha(\boldsymbol{a}) \doteq \max_{\boldsymbol{b} \in \mathcal{X}} \left\{ \psi(\boldsymbol{a}, \boldsymbol{b}) - \frac{\alpha}{2} \|\boldsymbol{a} - \boldsymbol{b}\|_2^2 \right\}$ and the associated proximal operator $\{\boldsymbol{b}^*(\boldsymbol{a})\} \doteq \arg\max_{\boldsymbol{b} \in \mathcal{X}} \left\{ \psi(\boldsymbol{a}, \boldsymbol{b}) - \frac{\alpha}{2} \|\boldsymbol{a} - \boldsymbol{b}\|_2^2 \right\}$, where $\psi$ is $\ell_{\nabla\psi}$-Lipschitz smooth. Then $\varphi_\alpha(\boldsymbol{a})$ is $\left( \ell_{\nabla\psi} + \frac{\ell_{\nabla\psi}^2}{\alpha} \right)$-Lipschitz-smooth, with gradients $\nabla_{\boldsymbol{a}} \varphi_\alpha(\boldsymbol{a}) = \nabla_{\boldsymbol{a}} \psi(\boldsymbol{a}, \boldsymbol{b}^*(\boldsymbol{a})) - \alpha(\boldsymbol{a} - \boldsymbol{b}^*(\boldsymbol{a}))$.*

*Proof of Theorem 4.1.* For notational clarity, let $\psi_\alpha(\boldsymbol{a}, \boldsymbol{b}) \doteq \psi(\boldsymbol{a}, \boldsymbol{b}) - \frac{\alpha}{2} \|\boldsymbol{a} - \boldsymbol{b}\|_2^2$. First, note that $\arg\max_{\boldsymbol{b} \in \mathcal{X}} \left\{ \psi(\boldsymbol{a}, \boldsymbol{b}) - \frac{\alpha}{2} \|\boldsymbol{a} - \boldsymbol{b}\|_2^2 \right\}$ is singleton-valued because it is the proximal operator associated with $\psi$ [101]. The differentiability of $\varphi_\alpha$ and its gradient then follow directly from Danskin's theorem [111] (or an envelope theorem [112, 113]). Let $\{\tilde{\boldsymbol{b}}\} \doteq \arg\max_{\boldsymbol{b} \in \mathcal{X}} \psi_\alpha(\tilde{\boldsymbol{a}}, \boldsymbol{b})$, $\{\hat{\boldsymbol{b}}\} \doteq \arg\max_{\boldsymbol{b} \in \mathcal{X}} \psi_\alpha(\hat{\boldsymbol{a}}, \boldsymbol{b})$. Now, notice that $\psi_\alpha$ is strongly-concave in $\boldsymbol{b}$ as it is the sum of $\psi$ which is concave in $\boldsymbol{b}$, and $\|\boldsymbol{a} - \boldsymbol{b}\|_2^2$ which is strongly concave in $\boldsymbol{b}$ [97]. Then, by the strong concavity of $\psi_\alpha$ in $\boldsymbol{b}$, we have:

$$\psi_\alpha(\tilde{\boldsymbol{a}}, \tilde{\boldsymbol{b}}) \leq \psi_\alpha(\tilde{\boldsymbol{a}}, \hat{\boldsymbol{b}}) + \left\langle \nabla_{\boldsymbol{b}} \psi_\alpha(\tilde{\boldsymbol{a}}, \hat{\boldsymbol{b}}), \tilde{\boldsymbol{b}} - \hat{\boldsymbol{b}} \right\rangle - \frac{\alpha}{2} \left\| \tilde{\boldsymbol{b}} - \hat{\boldsymbol{b}} \right\|_2^2 \tag{13}$$

$$\psi_\alpha(\tilde{\boldsymbol{a}}, \hat{\boldsymbol{b}}) \leq \psi_\alpha(\tilde{\boldsymbol{a}}, \tilde{\boldsymbol{b}}) + \underbrace{\left\langle \nabla_{\boldsymbol{b}} \psi_\alpha(\tilde{\boldsymbol{a}}, \tilde{\boldsymbol{b}}), \hat{\boldsymbol{b}} - \tilde{\boldsymbol{b}} \right\rangle}_{\leq 0 \text{ by FOC for } \tilde{\boldsymbol{b}}} - \frac{\alpha}{2} \left\| \tilde{\boldsymbol{b}} - \hat{\boldsymbol{b}} \right\|_2^2 \tag{14}$$

Summing up the two inequalities, we obtain:

$$\alpha \left\| \tilde{\boldsymbol{b}} - \hat{\boldsymbol{b}} \right\|_2^2 \leq \left\langle \nabla_{\boldsymbol{b}} \psi_\alpha(\tilde{\boldsymbol{a}}, \hat{\boldsymbol{b}}), \tilde{\boldsymbol{b}} - \hat{\boldsymbol{b}} \right\rangle \tag{15}$$

Additionally, since $\left\langle \nabla_{\boldsymbol{b}} \psi_\alpha(\hat{\boldsymbol{a}}, \hat{\boldsymbol{b}}), \tilde{\boldsymbol{b}} - \hat{\boldsymbol{b}} \right\rangle \leq 0$, we can substract it from the right hand side of the above inequality and obtain:

$$\alpha \left\| \tilde{\boldsymbol{b}} - \hat{\boldsymbol{b}} \right\|_2^2 \leq \left\langle \nabla_{\boldsymbol{b}} \psi_\alpha(\tilde{\boldsymbol{a}}, \hat{\boldsymbol{b}}), \tilde{\boldsymbol{b}} - \hat{\boldsymbol{b}} \right\rangle - \left\langle \nabla_{\boldsymbol{b}} \psi_\alpha(\hat{\boldsymbol{a}}, \hat{\boldsymbol{b}}), \tilde{\boldsymbol{b}} - \hat{\boldsymbol{b}} \right\rangle \tag{16}$$

$$= \left\langle \nabla_{\boldsymbol{b}} \psi_\alpha(\tilde{\boldsymbol{a}}, \hat{\boldsymbol{b}}) - \nabla_{\boldsymbol{b}} \psi_\alpha(\hat{\boldsymbol{a}}, \hat{\boldsymbol{b}}), \tilde{\boldsymbol{b}} - \hat{\boldsymbol{b}} \right\rangle \tag{17}$$

$$\leq \left\| \nabla_{\boldsymbol{b}} \psi_\alpha(\tilde{\boldsymbol{a}}, \hat{\boldsymbol{b}}) - \nabla_{\boldsymbol{b}} \psi_\alpha(\hat{\boldsymbol{a}}, \hat{\boldsymbol{b}}) \right\|_2 \left\| \tilde{\boldsymbol{b}} - \hat{\boldsymbol{b}} \right\|_2 \qquad \text{(Cauchy-Schwarz [114])} \tag{18}$$

$$\leq \ell_{\nabla \psi_\alpha} \left\| \tilde{\boldsymbol{a}} - \hat{\boldsymbol{a}} \right\|_2 \left\| \tilde{\boldsymbol{b}} - \hat{\boldsymbol{b}} \right\|_2 \qquad \text{(Lipschitz-smoothness of } \psi_\alpha) \tag{19}$$

Dividing both sides $\alpha \left\| \tilde{\boldsymbol{b}} - \hat{\boldsymbol{b}} \right\|_2$, we obtain:

$$\left\| \tilde{\boldsymbol{b}} - \hat{\boldsymbol{b}} \right\|_2 \leq \frac{\ell_{\nabla \psi_\alpha}}{\alpha} \left\| \tilde{\boldsymbol{a}} - \hat{\boldsymbol{a}} \right\|_2 \tag{20}$$

We now show that the parametric Moreau envelope of $\psi$, $\varphi_\alpha(\boldsymbol{a}) = \max_{\boldsymbol{b} \in \mathcal{X}} \psi_\alpha(\boldsymbol{a}, \boldsymbol{b})$, is Lipschitz-smooth. For any $\hat{\boldsymbol{a}}, \tilde{\boldsymbol{a}} \in \mathcal{A}$:

$$\left\| \nabla \varphi_\alpha(\hat{\boldsymbol{a}}) - \nabla \varphi_\alpha(\tilde{\boldsymbol{a}}) \right\|_2 \tag{21}$$

$$= \left\| \nabla_{\boldsymbol{b}} \psi_\alpha(\hat{\boldsymbol{a}}, \hat{\boldsymbol{b}}) - \nabla_{\boldsymbol{b}} \psi_\alpha(\tilde{\boldsymbol{a}}, \tilde{\boldsymbol{b}}) \right\|_2 \tag{22}$$

$$= \left\| \nabla_{\boldsymbol{b}} \psi_\alpha(\hat{\boldsymbol{a}}, \hat{\boldsymbol{b}}) - \nabla_{\boldsymbol{b}} \psi_\alpha(\tilde{\boldsymbol{a}}, \hat{\boldsymbol{b}}) + \nabla_{\boldsymbol{b}} \psi_\alpha(\tilde{\boldsymbol{a}}, \hat{\boldsymbol{b}}) - \nabla_{\boldsymbol{b}} \psi_\alpha(\tilde{\boldsymbol{a}}, \tilde{\boldsymbol{b}}) \right\|_2 \tag{23}$$

$$\leq \left\| \nabla_{\boldsymbol{b}} \psi_\alpha(\hat{\boldsymbol{a}}, \hat{\boldsymbol{b}}) - \nabla_{\boldsymbol{b}} \psi_\alpha(\tilde{\boldsymbol{a}}, \hat{\boldsymbol{b}}) \right\|_2 + \left\| \nabla_{\boldsymbol{b}} \psi_\alpha(\tilde{\boldsymbol{a}}, \hat{\boldsymbol{b}}) - \nabla_{\boldsymbol{b}} \psi_\alpha(\tilde{\boldsymbol{a}}, \tilde{\boldsymbol{b}}) \right\|_2 \tag{24}$$

$$\leq \ell_{\nabla \psi_\alpha} \left\| \hat{\boldsymbol{a}} - \tilde{\boldsymbol{a}} \right\|_2 + \ell_{\nabla \psi_\alpha} \left\| \hat{\boldsymbol{b}} - \tilde{\boldsymbol{b}} \right\|_2 \qquad \text{(Lipschitz-smoothness of } \psi_\alpha) \tag{25}$$

$$\leq \ell_{\nabla \psi_\alpha} \left\| \hat{\boldsymbol{a}} - \tilde{\boldsymbol{a}} \right\|_2 + \frac{\ell_{\nabla \psi_\alpha}^2}{\alpha} \left\| \hat{\boldsymbol{a}} - \tilde{\boldsymbol{a}} \right\|_2 \qquad \text{(Equation (20))} \tag{26}$$

$$\leq \left( \ell_{\nabla \psi_\alpha} + \frac{\ell_{\psi_\alpha}^2}{\alpha} \right) \left\| \tilde{\boldsymbol{a}} - \hat{\boldsymbol{a}} \right\|_2 \tag{27}$$

$\square$

Let $\psi_\alpha(\boldsymbol{a}, \boldsymbol{b}) = \psi(\boldsymbol{a}, \boldsymbol{b}) + \alpha/2 \left\| \boldsymbol{a} - \boldsymbol{b} \right\|_2^2$. The following theorem summarizes succinctly important properties of the $\alpha$-regularized-exploitability, $\varphi_\alpha(\boldsymbol{a}) = \max_{\boldsymbol{b} \in \mathcal{X}} \psi_\alpha(\boldsymbol{a}, \boldsymbol{b})$, which we use throughout this section:

**Theorem B.1.** *Given $\alpha > 0$, the following properties hold for $\alpha$-regularized exploitability and its associated solution correspondence $\mathcal{B}^*(\boldsymbol{a}) \doteq \arg\max_{\boldsymbol{b} \in \mathcal{X}} \psi_\alpha(\boldsymbol{a}, \boldsymbol{b})$:*

1. *$\psi_\alpha$ is $\alpha$ strongly concave in $\boldsymbol{b}$, and for all $\boldsymbol{a} \in \mathcal{A}$, the set $\mathcal{B}^*(\boldsymbol{a})$ is a singleton.*

2. *For any SE $(\boldsymbol{a}^*, \boldsymbol{b}^*)$ of $\min_{\boldsymbol{a} \in \mathcal{X}} \max_{\boldsymbol{b} \in \mathcal{X}} \psi_\alpha(\boldsymbol{a}, \boldsymbol{b})$, $\boldsymbol{a}^* = \boldsymbol{b}^*$. Additionally, $\boldsymbol{a}^*$ is a SE strategy of the outer player iff $\boldsymbol{a}^*$ is a VE of $\mathcal{G}$.*

3. *$\varphi_\alpha(\boldsymbol{a})$ is $\left( \ell_{\nabla \psi_\alpha} + \frac{\ell_{\nabla \psi_\alpha}^2}{\alpha} \right)$-Lipschitz-smooth with gradients given by for all $i \in [n]$:*

$$\nabla_{\boldsymbol{a}_i} \varphi_\alpha(\boldsymbol{a}) = \sum_{i \in [n]} \left[ \nabla_{\boldsymbol{a}_i} u_i(\boldsymbol{b}_i^*(\boldsymbol{a}), \boldsymbol{a}_{-i}) - \nabla_{\boldsymbol{a}_i} u_i(\boldsymbol{a}_i, \boldsymbol{a}_{-i}) \right] - \alpha(\boldsymbol{a}_i - \boldsymbol{b}_i^*(\boldsymbol{a}))$$

*where $\{\boldsymbol{b}^*(\boldsymbol{a})\} = \mathcal{B}^*(\boldsymbol{a})$*

*Proof.* (1) Notice that $\psi(\boldsymbol{a}, \boldsymbol{b})$ is concave in $\boldsymbol{b}$ for all $\boldsymbol{a} \in \mathcal{A}$, hence, since $\frac{\alpha}{2} \left\| \boldsymbol{a} - \boldsymbol{b} \right\|_2^2$ is $\alpha$-strongly convex in $\boldsymbol{b}$, $\psi_\alpha(\boldsymbol{a}, \boldsymbol{b})$ is strongly concave in $\boldsymbol{b}$, and $\arg\max_{\boldsymbol{b} \in \mathcal{X}} \psi_\alpha(\boldsymbol{a}, \boldsymbol{b})$ must be singleton-valued.

(2) First, note that a Stackelberg equilibrium of $\min_{\boldsymbol{a} \in \mathcal{X}} \max_{\boldsymbol{b} \in \mathcal{X}} \psi_\alpha(\boldsymbol{a}, \boldsymbol{b})$ is guaranteed to exist by continuity of $\psi_\alpha$ and compactness of the constraints.

(SE $\Leftarrow$ VE): Suppose that $\boldsymbol{a}^* \in \mathcal{X}$ is a VE of $\mathcal{G}$, then for all $\boldsymbol{b} \in \mathcal{A}$ we have:

$$\psi_\alpha(\boldsymbol{a}^*, \boldsymbol{b}) = \psi(\boldsymbol{a}^*, \boldsymbol{b}) - \frac{\alpha}{2} \|\boldsymbol{a}^* - \boldsymbol{b}\|_2^2 \tag{28}$$

$$= \sum_{i \in [n]} \left[ \underbrace{u_i(\boldsymbol{b}_i, \boldsymbol{a}_{-i}) - u_i(\boldsymbol{a}_i, \boldsymbol{a}_{-i})}_{\leq 0 \text{ by GNE definition}} \right] - \frac{\alpha}{2} \|\boldsymbol{a}^* - \boldsymbol{b}\|_2^2 \tag{29}$$

$$\leq -\frac{\alpha}{2} \|\boldsymbol{a}^* - \boldsymbol{b}\|_2^2 \tag{30}$$

Taking the max over $\boldsymbol{b}$ on both sides of the final inequality, we obtain

$$\max_{\boldsymbol{b} \in \mathcal{X}} \psi_\alpha(\boldsymbol{a}^*, \boldsymbol{b}) \leq \max_{\boldsymbol{b} \in \mathcal{X}} -\frac{\alpha}{2} \|\boldsymbol{a}^* - \boldsymbol{b}\|_2^2 \leq 0 \ .$$

Note that for all $\boldsymbol{a} \in \mathcal{X}$, $\max_{\boldsymbol{b} \in \mathcal{X}} \psi_\alpha(\boldsymbol{a}, \boldsymbol{b}) \geq \psi_\alpha(\boldsymbol{a}, \boldsymbol{a}) = 0$, hence we must have that $\boldsymbol{a}^*$ is the SE strategy of the outer player and the SE strategy of the inner player $\boldsymbol{b}^*$ must be equal to $\boldsymbol{a}^*$.

(SE $\Rightarrow$ VE): Let $(\boldsymbol{a}^*, \boldsymbol{b}^*)$ be an SE of $\min_{\boldsymbol{a} \in \mathcal{X}} \max_{\boldsymbol{b} \in \mathcal{X}} \psi_\alpha(\boldsymbol{a}, \boldsymbol{b})$. Note that for all $\boldsymbol{a} \in \mathcal{X}$, $\max_{\boldsymbol{b} \in \mathcal{X}} \psi_\alpha(\boldsymbol{a}, \boldsymbol{b}) \geq \psi_\alpha(\boldsymbol{a}, \boldsymbol{a}) = 0$. Since under our assumptions a GNE is guaranteed to exist, there exists $\boldsymbol{a}'$ such that $\max_{\boldsymbol{b} \in \mathcal{A}} \psi_\alpha(\boldsymbol{a}', \boldsymbol{b}) = 0$, we must then have that $\min_{\boldsymbol{a} \in \mathcal{X}} \max_{\boldsymbol{b} \in \mathcal{A}} \psi_\alpha(\boldsymbol{a}, \boldsymbol{b}) \leq 0$, i.e.:

$$\max_{\boldsymbol{b} \in \mathcal{X}} \left\{ \sum_{i \in [n]} \left[ u_i(\boldsymbol{b}_i, \boldsymbol{a}_{-i}^*) - u_i(\boldsymbol{a}_i^*, \boldsymbol{a}_{-i}^*) \right] - \frac{\alpha}{2} \|\boldsymbol{a}^* - \boldsymbol{b}\|_2^2 \right\} \leq 0 \tag{31}$$

We will show that the exploitability of $\boldsymbol{a}^*$ is zero w.r.t. the pseudo-game $\mathcal{G}$, which will imply that $\boldsymbol{a}^*$ is a GNE. First, by the above equation, we have for any $\boldsymbol{b} \in \mathcal{A}$:

$$\sum_{i \in [n]} \left[ u_i(\boldsymbol{b}_i, \boldsymbol{a}_{-i}^*) - u_i(\boldsymbol{a}_i^*, \boldsymbol{a}_{-i}^*) \right] - \frac{\alpha}{2} \|\boldsymbol{a}^* - \boldsymbol{b}\|_2^2 \leq 0 \tag{32}$$

For $\lambda \in (0, 1)$, and any $\boldsymbol{a} \in \mathcal{A}$, let $\boldsymbol{b}_i = \lambda \boldsymbol{a}_i^* + (1 - \lambda)\boldsymbol{a}_i$ in the above inequality, we then have:

$$0 \geq \sum_{i \in [n]} \left[ u_i(\lambda \boldsymbol{a}_i^* + (1 - \lambda)\boldsymbol{a}_i, \boldsymbol{a}_{-i}^*) - u_i(\boldsymbol{a}_i^*, \boldsymbol{a}_{-i}^*) \right] - \frac{\alpha}{2} \|\boldsymbol{a}^* - [\lambda \boldsymbol{a}^* + (1 - \lambda)\boldsymbol{a}]\|_2^2 \tag{33}$$

$$= \sum_{i \in [n]} \left[ u_i(\lambda \boldsymbol{a}_i^* + (1 - \lambda)\boldsymbol{a}_i, \boldsymbol{a}_{-i}^*) - u_i(\boldsymbol{a}_i^*, \boldsymbol{a}_{-i}^*) \right] - \frac{\alpha}{2} \|(1 - \lambda)\boldsymbol{a}^* - (1 - \lambda)\boldsymbol{a}\|_2^2 \tag{34}$$

$$= \sum_{i \in [n]} \left[ u_i(\lambda \boldsymbol{a}_i^* + (1 - \lambda)\boldsymbol{a}_i, \boldsymbol{a}_{-i}^*) - u_i(\boldsymbol{a}_i^*, \boldsymbol{a}_{-i}^*) \right] - \frac{\alpha}{2}(1 - \lambda)^2 \|\boldsymbol{a}^* - \boldsymbol{a}\|_2^2 \tag{35}$$

$$\geq \sum_{i \in [n]} \left[ \lambda u_i(\boldsymbol{a}_i^*, \boldsymbol{a}_{-i}^*) + (1 - \lambda)u_i(\boldsymbol{a}_i, \boldsymbol{a}_{-i}^*) - u_i(\boldsymbol{a}_i^*, \boldsymbol{a}_{-i}^*) \right] - \frac{\alpha}{2}(1 - \lambda)^2 \|\boldsymbol{a}^* - \boldsymbol{a}_i\|_2^2 \tag{36}$$

$$\tag{37}$$

Taking $\lambda \to 1^-$, by continuity of the $l_2$-norm and the utility functions, we have:

$$0 \geq \sum_{i \in [n]} \left[ u_i(\boldsymbol{a}_i^*, \boldsymbol{a}_{-i}^*) - u_i(\boldsymbol{a}_i^*, \boldsymbol{a}_{-i}^*) \right] \tag{38}$$

which implies that $\boldsymbol{a}^*$ is a GNE, additionally for $\sum_{i \in [n]} \left[ u_i(\boldsymbol{b}_i, \boldsymbol{a}_{-i}^*) - u_i(\boldsymbol{a}_i^*, \boldsymbol{a}_{-i}^*) \right] - \frac{\alpha}{2} \|\boldsymbol{a}^* - \boldsymbol{b}\|_2^2 \leq 0$ to be minimized by $\boldsymbol{a}^*$, we must have that $\boldsymbol{b} = \boldsymbol{a}^*$.

(3): Follows from Theorem 4.1.

$\square$

Before we move forward, we introduce the following definitions. A function $f : \mathcal{A} \to \mathbb{R}$ is said to be $\mu$-**Polyak-Lojasiewicz (PL)** if for all $\boldsymbol{x} \in \mathcal{X}$, $1/2 \left\| \nabla f(\boldsymbol{x}) \right\|_2^2 \geq \mu(f(\boldsymbol{x}) - \min_{\boldsymbol{x} \in \mathcal{X}} f(\boldsymbol{x}))$. A function $f : \mathcal{A} \to \mathbb{R}$ is said to be $\mu$-**quadratically growing (QG)**, if for all $\boldsymbol{x} \in \mathcal{X}$, $f(\boldsymbol{x}) - \min_{\boldsymbol{x} \in \mathcal{X}} f(\boldsymbol{x}) \geq \mu/2 \left\| \boldsymbol{x}^* - \boldsymbol{x} \right\|^2$ where $\boldsymbol{x}^* \in \arg\min_{\boldsymbol{x} \in \mathcal{X}} f(\boldsymbol{x})$. We note that any $\mu$-SC function is PL (Appendix A, [103]), and that any $\mu$-PL function is $4\mu$-QG, we restate the following lemma for convenience as we will use it in the subsequent proofs:

**Lemma B.2** (Corollary of Theorem 2 [103]). *If a function $f$ satisfies is $\mu$-PL, then $f$ is $4\mu$-quadratically-growing.*

In order to obtain our convergence rates, we first bound the error between the true gradient of $\varphi_\alpha$ and its approximation by ADA:

**Lemma B.3** (Inner Loop Error Bound). *Let $\varphi_\alpha(\boldsymbol{a}) = \max_{\boldsymbol{b} \in \mathcal{A}} \psi_\alpha(\boldsymbol{a}, \boldsymbol{b})$, let $c = \max_{(\boldsymbol{a},\boldsymbol{b}) \in \mathcal{X} \times \mathcal{X}} (\psi_\alpha(\boldsymbol{a}, \boldsymbol{b}) - \psi_\alpha(\boldsymbol{a}, \boldsymbol{0}))$, and define $\nabla_{\boldsymbol{a}} \varphi_\alpha$ as in Theorem B.1. Suppose that ADA is run on a pseudo-game $\mathcal{G}$ which satisfies Assumption 3.3 with learning rates $\eta_{\boldsymbol{a}} > 0$ and $\eta_{\boldsymbol{b}} = \frac{1}{\ell_{\psi_\alpha}}$, for any number of outer loop iterations $T_{\boldsymbol{a}} \in \mathbb{N}_{++}$, and for $T_{\boldsymbol{b}} \geq \frac{2 \log\left( \frac{\varepsilon}{\ell_{\nabla \psi_\alpha}} \sqrt{\frac{2\alpha}{c}} \right)}{\log\left( \frac{\alpha}{\ell_{\nabla \psi_\alpha}} \right)}$ total inner loop iterations, where $\varepsilon > 0$. Then, the outputs $(\boldsymbol{a}^{(t)}, \boldsymbol{b}^{(t)})_{t=1}^{T_{\boldsymbol{a}}}$ satisfy $\left\| \nabla_{\boldsymbol{a}} \varphi_\alpha(\boldsymbol{a}^{(t)}) - \nabla_{\boldsymbol{a}} \psi_\alpha(\boldsymbol{a}^{(t)}, \boldsymbol{b}^{(t)}) \right\| \leq \varepsilon$.*

*Proof.* Note that $\arg\max_{\boldsymbol{b} \in \mathcal{A}} \psi_\alpha(\boldsymbol{a}, \boldsymbol{b})$ is singleton-valued by Theorem B.1. Let $\{\boldsymbol{b}^*(\boldsymbol{a})\} = \arg\max_{\boldsymbol{b} \in \mathcal{A}} \psi_\alpha(\boldsymbol{a}, \boldsymbol{b})$.

Since $\psi_\alpha$ is $\alpha$-strongly-concave in $\boldsymbol{b}$ and $\ell_{\nabla \psi_\alpha}$-Lipschitz-smooth (Theorem B.1), we have from Theorem 1 of Karimi, Nutini, and Schmidt [103]:

$$\varphi_\alpha(\boldsymbol{a}^{(t)}) - \psi_\alpha(\boldsymbol{a}^{(t)}, \boldsymbol{b}^{(t)}) \leq \left( 1 - \frac{\alpha}{\ell_{\nabla \psi_\alpha}} \right)^t \left( \varphi_\alpha(\boldsymbol{a}^{(t)}) - \psi_\alpha(\boldsymbol{a}^{(t)}, \boldsymbol{0}) \right) \tag{39}$$

Then, since $\psi_\alpha$ is $\alpha$-strongly-convex, by Lemma B.2, we have:

$$\varphi_\alpha(\boldsymbol{a}^{(t)}) - \psi_\alpha(\boldsymbol{a}^{(t)}, \boldsymbol{b}^{(t)}) \geq 2\alpha \left\| \boldsymbol{b}^*(\boldsymbol{a}^{(t)}) - \boldsymbol{b}^{(t)} \right\|_2^2 \;\;, \tag{40}$$

Combining the two previous inequalities, we get:

$$\left\| \boldsymbol{b}^*(\boldsymbol{a}^{(t)}) - \boldsymbol{b}^{(t)} \right\|_2 \leq \left( \frac{\alpha}{\ell_{\nabla \psi_\alpha}} \right)^{\frac{t}{2}} \sqrt{\frac{(\varphi_\alpha(\boldsymbol{a}^{(t)}) - \psi_\alpha(\boldsymbol{a}^{(t)}, \boldsymbol{0}))}{2\alpha}} \tag{41}$$

Finally, we bound the error between the approximate gradient computed by ADA $\nabla \psi_{\boldsymbol{a}}(\boldsymbol{a}^{(t)}, \boldsymbol{b}^{(t)})$ and the true gradient $\nabla \varphi_\alpha(\boldsymbol{a}^{(t)})$ at each iteration $t \in \mathbb{N}_{++}$:

$$\left\| \nabla \varphi_\alpha(\boldsymbol{a}^{(t)}) - \nabla \psi_{\boldsymbol{a}}(\boldsymbol{a}^{(t)}, \boldsymbol{b}^{(t)}) \right\|_2 = \left\| \nabla \psi(\boldsymbol{a}^{(t)}, \boldsymbol{b}^*(\boldsymbol{a}^{(t)})) - \nabla_{\boldsymbol{a}} \psi(\boldsymbol{a}^{(t)}, \boldsymbol{b}^{(t)}) \right\|_2 \tag{42}$$

$$\leq \ell_{\nabla \psi_\alpha} \left\| (\boldsymbol{a}^{(t)}, \boldsymbol{b}^*(\boldsymbol{a}^{(t)})) - (\boldsymbol{a}^{(t)}, \boldsymbol{b}^{(t)}) \right\|_2 \tag{43}$$

$$\leq \ell_{\nabla \psi_\alpha} \left( \left\| \boldsymbol{a}^{(t)} - \boldsymbol{a}^{(t)} \right\|_2 + \left\| \boldsymbol{b}^*(\boldsymbol{a}^{(t)}) - \boldsymbol{b}^{(t)} \right\|_2 \right) \tag{44}$$

$$= \ell_{\nabla \psi_\alpha} \left\| \boldsymbol{b}^*(\boldsymbol{a}^{(t)}) - \boldsymbol{b}^{(t)} \right\|_2 \tag{45}$$

$$\leq \ell_{\nabla \psi_\alpha} \left( \frac{\alpha}{\ell_{\nabla \psi_\alpha}} \right)^{\frac{t}{2}} \sqrt{\frac{(\varphi_\alpha(\boldsymbol{a}^{(t)}) - \psi_\alpha(\boldsymbol{a}^{(t)}, \boldsymbol{0}))}{2\alpha}} \qquad \text{(Equation (41))} \tag{46}$$

$$\leq \ell_{\nabla \psi_\alpha} \left( \frac{\alpha}{\ell_{\nabla \psi_\alpha}} \right)^{\frac{t}{2}} \sqrt{\frac{\max_{(\boldsymbol{a},\boldsymbol{b}) \in \mathcal{X} \times \mathcal{X}} (\psi_\alpha(\boldsymbol{a}, \boldsymbol{b}) - \psi_\alpha(\boldsymbol{a}, \boldsymbol{0}))}{2\alpha}} \tag{47}$$

Let $c = \max_{(\boldsymbol{a},\boldsymbol{b}) \in \mathcal{X} \times \mathcal{X}} (\psi_\alpha(\boldsymbol{a}, \boldsymbol{b}) - \psi_\alpha(\boldsymbol{a}, \boldsymbol{0}))$, we obtain:

$$\left\| \nabla \varphi_\alpha(\boldsymbol{a}^{(t)}) - \nabla \psi_{\boldsymbol{a}}(\boldsymbol{a}^{(t)}, \boldsymbol{b}^{(t)}) \right\| \le \ell_{\nabla \psi_\alpha} \left( \frac{\alpha}{\ell_{\nabla \psi_\alpha}} \right)^{\frac{t}{2}} \sqrt{\frac{c}{2\alpha}} \tag{48}$$

Then, given $\varepsilon > 0$, for any number of inner loop iterations such that $T_{\boldsymbol{b}} \ge \frac{2\log\left( \frac{\varepsilon}{\ell_{\nabla \psi_\alpha}} \sqrt{\frac{2\alpha}{c}} \right)}{\log\left( \frac{\alpha}{\ell_{\nabla \psi_\alpha}} \right)}$, for all $t \in [T_{\boldsymbol{a}}]$, we have:

$$\left\| \nabla \varphi_\alpha(\boldsymbol{a}^{(t)}) - \nabla \psi_{\boldsymbol{a}}(\boldsymbol{a}^{(t)}, \boldsymbol{b}^{(t)}) \right\| \le \varepsilon \tag{49}$$

$\square$

**Lemma B.4** (Progress Lemma for Approximate Iterate). *Suppose that ADA is run on a pseudo-game $\mathcal{G}$ which satisfies Assumption 3.3 with learning rates $\eta_{\boldsymbol{a}} > 0$ and $\eta_{\boldsymbol{b}} = \frac{1}{\ell_{\nabla \psi_\alpha}}$, for any number of outer loop iterations $T_{\boldsymbol{a}} \in \mathbb{N}_{++}$ and for $T_{\boldsymbol{b}} \ge \frac{2\log\left( \frac{\varepsilon}{\ell_{\nabla \psi_\alpha}} \sqrt{\frac{2\alpha}{c}} \right)}{\log\left( \frac{\alpha}{\ell_{\nabla \psi_\alpha}} \right)}$ total inner loop iterations where $\varepsilon > 0$. Then the outputs $(\boldsymbol{a}^{(t)}, \boldsymbol{b}^{(t)})_{t=0}^{T_{\boldsymbol{a}}}$ satisfy $\varphi_\alpha(\boldsymbol{a}^{(t+1)}) - \varphi_\alpha(\boldsymbol{a}^{(t)}) \le \left( \frac{\ell_{\nabla \psi_\alpha} + \frac{\ell_{\nabla \psi_\alpha}^2}{\alpha}}{2} - \frac{1}{\eta_{\boldsymbol{a}}} \right) \left\| G_{\eta_{\boldsymbol{a}}}^{\varphi_\alpha}(\boldsymbol{a}^{(t)}) \right\|_2^2 + \left( (2\ell_{\varphi_\alpha} + \varepsilon) \left( \frac{\ell_{\nabla \psi_\alpha} + \frac{\ell_{\nabla \psi_\alpha}^2}{\alpha}}{2} - \frac{1}{\eta_{\boldsymbol{a}}} \right) + \ell_{\psi_\alpha} \right) \varepsilon$*

*Proof of Lemma B.4.* Note that by the second projection/proximal theorem [115, 101], we have for all $\boldsymbol{a} \in \mathcal{A}$:

$$\left\langle \boldsymbol{a}^{(t)} - \eta_{\boldsymbol{a}} \nabla_{\boldsymbol{a}} \psi_\alpha(\boldsymbol{a}^{(t)}, \boldsymbol{b}^{(t)}) - \boldsymbol{a}^{(t+1)}, \boldsymbol{a} - \boldsymbol{a}^{(t+1)} \right\rangle \le 0 \tag{50}$$

$$\left\langle \nabla_{\boldsymbol{a}} \psi_\alpha(\boldsymbol{a}^{(t)}, \boldsymbol{b}^{(t)}), \boldsymbol{a} - \boldsymbol{a}^{(t+1)} \right\rangle \ge \frac{1}{\eta_{\boldsymbol{a}}} \left\langle \boldsymbol{a}^{(t)} - \boldsymbol{a}^{(t+1)}, \boldsymbol{a} - \boldsymbol{a}^{(t+1)} \right\rangle \tag{51}$$

by setting $\boldsymbol{a} = \boldsymbol{a}^{(t)}$, and applying Cauchy-Schwarz [115] to the right hand side of the above inequality, it follows that:

$$\left\langle \nabla \psi_\alpha(\boldsymbol{a}^{(t)}, \boldsymbol{b}^{(t)}), \boldsymbol{a}^{(t+1)} - \boldsymbol{a}^{(t)} \right\rangle \le -\frac{1}{\eta_{\boldsymbol{a}}} \left\| \boldsymbol{a}^{(t+1)} - \boldsymbol{a}^{(t)} \right\|_2^2 \tag{52}$$

$$\left\langle \nabla \varphi_\alpha(\boldsymbol{a}^{(t)}), \boldsymbol{a}^{(t+1)} - \boldsymbol{a}^{(t)} \right\rangle \le -\frac{1}{\eta_{\boldsymbol{a}}} \left\| \boldsymbol{a}^{(t+1)} - \boldsymbol{a}^{(t)} \right\|_2^2 + \left\langle \nabla \varphi_\alpha(\boldsymbol{a}^{(t)}) - \nabla_{\boldsymbol{a}} \psi_\alpha(\boldsymbol{a}^{(t)}, \boldsymbol{b}^{(t)}), \boldsymbol{a}^{(t+1)} - \boldsymbol{a}^{(t)} \right\rangle \tag{53}$$

$$\left\langle \nabla \varphi_\alpha(\boldsymbol{a}^{(t)}), \boldsymbol{a}^{(t+1)} - \boldsymbol{a}^{(t)} \right\rangle \le -\frac{1}{\eta_{\boldsymbol{a}}} \left\| \boldsymbol{a}^{(t+1)} - \boldsymbol{a}^{(t)} \right\|_2^2 + \left\langle \nabla \varphi_\alpha(\boldsymbol{a}^{(t)}) - \nabla_{\boldsymbol{a}} \psi_\alpha(\boldsymbol{a}^{(t)}, \boldsymbol{b}^{(t)}), \boldsymbol{a}^{(t+1)} - \boldsymbol{a}^{(t)} \right\rangle \tag{54}$$

Define $\mathrm{err}^{(t)} \doteq \nabla_{\boldsymbol{a}} \varphi_\alpha(\boldsymbol{a}^{(t)}) - \nabla \psi_\alpha(\boldsymbol{a}^{(t)}, \boldsymbol{b}^{(t)})$, we get:

$$\left\langle \nabla \varphi_\alpha(\boldsymbol{a}^{(t)}), \boldsymbol{a}^{(t+1)} - \boldsymbol{a}^{(t)} \right\rangle \le -\frac{1}{\eta_{\boldsymbol{a}}} \left\| \boldsymbol{a}^{(t+1)} - \boldsymbol{a}^{(t)} \right\|_2^2 + \left\langle \mathrm{err}^{(t)}, \boldsymbol{a}^{(t+1)} - \boldsymbol{a}^{(t)} \right\rangle \tag{55}$$

$$\le -\frac{1}{\eta_{\boldsymbol{a}}} \left\| \boldsymbol{a}^{(t+1)} - \boldsymbol{a}^{(t)} \right\|_2^2 + \left\| \mathrm{err}^{(t)} \right\| \left\| \boldsymbol{a}^{(t+1)} - \boldsymbol{a}^{(t)} \right\| \tag{56}$$

Let $\varphi_\alpha(\boldsymbol{a}) = \max_{\boldsymbol{b} \in \mathcal{A}} \psi_\alpha(\boldsymbol{a}, \boldsymbol{b})$ and $\nabla_{\boldsymbol{a}} \varphi_\alpha$ as in Theorem B.1. By Theorem B.1, we have that $\varphi_\alpha$ is $\left( \ell_{\nabla \psi_\alpha} + \frac{\ell_{\nabla \psi_\alpha}^2}{\alpha} \right)$-Lipschitz-smooth. Hence, for all $t \in [T_{\boldsymbol{a}}]$:

$$\varphi_\alpha(\boldsymbol{a}^{(t+1)}) \le \varphi_\alpha(\boldsymbol{a}^{(t)}) + \left\langle \nabla_{\boldsymbol{a}} \varphi_\alpha(\boldsymbol{a}^{(t)}), \boldsymbol{a}^{(t+1)} - \boldsymbol{a}^{(t)} \right\rangle + \frac{\ell_{\nabla \psi_\alpha} + \frac{\ell_{\nabla \psi_\alpha}^2}{\alpha}}{2} \left\| \boldsymbol{a}^{(t+1)} - \boldsymbol{a}^{(t)} \right\|_2^2 \tag{57}$$

which combined with Equation (56), yields:

$$\varphi_\alpha(\boldsymbol{a}^{(t+1)}) \leq \varphi_\alpha(\boldsymbol{a}^{(t)}) - \frac{1}{\eta_{\boldsymbol{a}}} \left\| \boldsymbol{a}^{(t+1)} - \boldsymbol{a}^{(t)} \right\|_2^2 + \left\| \mathrm{err}^{(t)} \right\| \left\| \boldsymbol{a}^{(t+1)} - \boldsymbol{a}^{(t)} \right\| + \frac{\ell_{\nabla \psi_\alpha} + \frac{\ell_{\nabla \psi_\alpha}^2}{\alpha}}{2} \left\| \boldsymbol{a}^{(t+1)} - \boldsymbol{a}^{(t)} \right\|_2^2$$
(58)

$$\leq \varphi_\alpha(\boldsymbol{a}^{(t)}) + \left( \frac{\ell_{\nabla \psi_\alpha} + \frac{\ell_{\nabla \psi_\alpha}^2}{\alpha}}{2} - \frac{1}{\eta_{\boldsymbol{a}}} \right) \left\| \boldsymbol{a}^{(t+1)} - \boldsymbol{a}^{(t)} \right\|_2^2 + \left\| \mathrm{err}^{(t)} \right\| \left\| \boldsymbol{a}^{(t+1)} - \boldsymbol{a}^{(t)} \right\|$$
(59)

$$\leq \varphi_\alpha(\boldsymbol{a}^{(t)}) + \left( \frac{\ell_{\nabla \psi_\alpha} + \frac{\ell_{\nabla \psi_\alpha}^2}{\alpha}}{2} - \frac{1}{\eta_{\boldsymbol{a}}} \right) \left\| \boldsymbol{a}^{(t+1)} - \boldsymbol{a}^{(t)} \right\|_2^2 + \left\| \mathrm{err}^{(t)} \right\| \left\| \boldsymbol{a}^{(t+1)} - \boldsymbol{a}^{(t)} \right\|$$
(60)

Re-organizing expressions, we obtain:

$$\varphi_\alpha(\boldsymbol{a}^{(t+1)}) - \varphi_\alpha(\boldsymbol{a}^{(t)}) \leq \left( \frac{\ell_{\nabla \psi_\alpha} + \frac{\ell_{\nabla \psi_\alpha}^2}{\alpha}}{2} - \frac{1}{\eta_{\boldsymbol{a}}} \right) \left\| \boldsymbol{a}^{(t+1)} - \boldsymbol{a}^{(t)} \right\|_2^2 + \left\| \mathrm{err}^{(t)} \right\| \left\| \boldsymbol{a}^{(t+1)} - \boldsymbol{a}^{(t)} \right\|$$
(61)

$$\leq \left( \frac{\ell_{\nabla \psi_\alpha} + \frac{\ell_{\nabla \psi_\alpha}^2}{\alpha}}{2} - \frac{1}{\eta_{\boldsymbol{a}}} \right) \left\| G_{\eta_{\boldsymbol{a}}}^{\psi_\alpha(\cdot, \boldsymbol{b}^{(t)})}(\boldsymbol{a}^{(t)}) \right\|_2^2 + \left\| \mathrm{err}^{(t)} \right\| \left\| G_{\eta_{\boldsymbol{a}}}^{\psi_\alpha(\cdot, \boldsymbol{b}^{(t)})}(\boldsymbol{a}^{(t)}) \right\|$$
(62)

$$\leq \left( \frac{\ell_{\nabla \psi_\alpha} + \frac{\ell_{\nabla \psi_\alpha}^2}{\alpha}}{2} - \frac{1}{\eta_{\boldsymbol{a}}} \right) \left\| G_{\eta_{\boldsymbol{a}}}^{\psi_\alpha(\cdot, \boldsymbol{b}^{(t)})}(\boldsymbol{a}^{(t)}) \right\|_2^2 + \ell_{\psi_\alpha} \left\| \mathrm{err}^{(t)} \right\|$$
(63)

where the last line follows from the fact that $\left\| G_{\eta_{\boldsymbol{a}}}^{\psi_\alpha(\cdot, \boldsymbol{b}^{(t)})}(\boldsymbol{a}) \right\|_2 \leq \left\| \nabla_{\boldsymbol{a}} \psi_\alpha(\boldsymbol{a}) \right\|$ (Proposition 2.4, [116]) and the Lipschitz-smoothness of $\psi_\alpha$. Additionally, note that we have:

$$\left\| G_{\eta_{\boldsymbol{a}}}^{\psi_\alpha(\cdot, \boldsymbol{b}^{(t)})}(\boldsymbol{a}^{(t)}) \right\|_2 = \left\| \Pi_{\mathcal{X}} \left[ \boldsymbol{a}^{(t)} - \eta_{\boldsymbol{a}} \nabla_{\boldsymbol{a}} \psi(\boldsymbol{a}^{(t)}, \boldsymbol{b}^{(t)}) \right] - \boldsymbol{a}^{(t)} \right\|_2$$
(64)

$$= \left\| \Pi_{\mathcal{X}} \left[ \boldsymbol{a}^{(t)} - \eta_{\boldsymbol{a}} \left( \nabla_{\boldsymbol{a}} \varphi_\alpha(\boldsymbol{a}^{(t)}) + \nabla_{\boldsymbol{a}} \psi(\boldsymbol{a}^{(t)}, \boldsymbol{b}^{(t)}) - \nabla_{\boldsymbol{a}} \varphi_\alpha(\boldsymbol{a}^{(t)}) \right) \right] - \boldsymbol{a}^{(t)} \right\|_2$$
(65)

$$\leq \left\| \Pi_{\mathcal{X}} \left[ \boldsymbol{a}^{(t)} - \eta_{\boldsymbol{a}} \left( \nabla_{\boldsymbol{a}} \varphi_\alpha(\boldsymbol{a}^{(t)}) \right) \right] - \boldsymbol{a}^{(t)} \right\|_2 + \left\| \nabla_{\boldsymbol{a}} \psi(\boldsymbol{a}^{(t)}, \boldsymbol{b}^{(t)}) - \nabla_{\boldsymbol{a}} \varphi_\alpha(\boldsymbol{a}^{(t)}) \right\|_2$$
(66)

$$\leq \left\| G_{\eta_{\boldsymbol{a}}}^{\varphi_\alpha}(\boldsymbol{a}^{(t)}) \right\|_2 + \left\| \mathrm{err}^{(t)} \right\|$$
(67)

where the penultimate line follows from Proposition 2.4 of Hazan, Singh, and Zhang [116]. Going back to Equation (63), we have:

$$\varphi_\alpha(\boldsymbol{a}^{(t+1)}) - \varphi_\alpha(\boldsymbol{a}^{(t)}) \tag{68}$$

$$\leq \left( \frac{\ell_{\nabla \psi_\alpha} + \frac{\ell_{\nabla \psi_\alpha}^2}{\alpha}}{2} - \frac{1}{\eta_{\boldsymbol{a}}} \right) \left( \left\| G_{\eta_{\boldsymbol{a}}}^{\varphi_\alpha}(\boldsymbol{a}^{(t)}) \right\|_2 + \left\| \mathrm{err}^{(t)} \right\| \right)^2 + \ell_{\psi_\alpha} \left\| \mathrm{err}^{(t)} \right\| \tag{69}$$

$$\leq \left( \frac{\ell_{\nabla \psi_\alpha} + \frac{\ell_{\nabla \psi_\alpha}^2}{\alpha}}{2} - \frac{1}{\eta_{\boldsymbol{a}}} \right) \left( \left\| G_{\eta_{\boldsymbol{a}}}^{\varphi_\alpha}(\boldsymbol{a}^{(t)}) \right\|_2^2 + 2 \left\| G_{\eta_{\boldsymbol{a}}}^{\varphi_\alpha}(\boldsymbol{a}^{(t)}) \right\|_2 \left\| \mathrm{err}^{(t)} \right\| + \left\| \mathrm{err}^{(t)} \right\|_2^2 \right) + \ell_{\psi_\alpha} \left\| \mathrm{err}^{(t)} \right\|$$

$$\tag{70}$$

$$\leq \left( \frac{\ell_{\nabla \psi_\alpha} + \frac{\ell_{\nabla \psi_\alpha}^2}{\alpha}}{2} - \frac{1}{\eta_{\boldsymbol{a}}} \right) \left( \left\| G_{\eta_{\boldsymbol{a}}}^{\varphi_\alpha}(\boldsymbol{a}^{(t)}) \right\|_2^2 + 2\ell_{\varphi_\alpha} \left\| \mathrm{err}^{(t)} \right\| + \left\| \mathrm{err}^{(t)} \right\|_2^2 \right) + \ell_{\psi_\alpha} \left\| \mathrm{err}^{(t)} \right\| \tag{71}$$

$$\leq \left( \frac{\ell_{\nabla \psi_\alpha} + \frac{\ell_{\nabla \psi_\alpha}^2}{\alpha}}{2} - \frac{1}{\eta_{\boldsymbol{a}}} \right) \left\| G_{\eta_{\boldsymbol{a}}}^{\varphi_\alpha}(\boldsymbol{a}^{(t)}) \right\|_2^2 + \left( \left( 2\ell_{\varphi_\alpha} + \left\| \mathrm{err}^{(t)} \right\| \right) \left( \frac{\ell_{\nabla \psi_\alpha} + \frac{\ell_{\nabla \psi_\alpha}^2}{\alpha}}{2} - \frac{1}{\eta_{\boldsymbol{a}}} \right) + \ell_{\psi_\alpha} \right) \left\| \mathrm{err}^{(t)} \right\|$$

$$\tag{72}$$

The conditions of Lemma B.3 are satisfied by our lemma statement and hence we have for all $t \in [T_{\boldsymbol{a}}]$, $\left\| \mathrm{err}^{(t)} \right\| \leq \varepsilon$, giving us:

$$\varphi_\alpha(\boldsymbol{a}^{(t+1)}) - \varphi_\alpha(\boldsymbol{a}^{(t)}) \leq \left( \frac{\ell_{\nabla \psi_\alpha} + \frac{\ell_{\nabla \psi_\alpha}^2}{\alpha}}{2} - \frac{1}{\eta_{\boldsymbol{a}}} \right) \left\| G_{\eta_{\boldsymbol{a}}}^{\varphi_\alpha}(\boldsymbol{a}^{(t)}) \right\|_2^2 + \left( (2\ell_{\varphi_\alpha} + \varepsilon) \left( \frac{\ell_{\nabla \psi_\alpha} + \frac{\ell_{\nabla \psi_\alpha}^2}{\alpha}}{2} - \frac{1}{\eta_{\boldsymbol{a}}} \right) + \ell_{\psi_\alpha} \right) \varepsilon$$

$$\tag{73}$$

$\square$

**Theorem 4.2** (Convergence to Stationary Point of Exploitability). *Suppose that ADA is run on a pseudo-game $\mathcal{G}$ which satisfies Assumption 3.3 with learning rates $\eta_{\boldsymbol{a}} > \dfrac{2}{\ell_{\nabla \psi_\alpha} + \frac{\ell_{\nabla \psi_\alpha}^2}{\alpha}}$ and $\eta_{\boldsymbol{b}} = \dfrac{1}{\ell_{\psi_\alpha}}$,*

*for any number of outer loop iterations $T_{\boldsymbol{a}} \in \mathbb{N}_{++}$ and for $T_{\boldsymbol{b}} \geq \dfrac{2\log\left( \frac{\varepsilon}{\ell_{\nabla \psi_\alpha}} \sqrt{\frac{2\alpha}{c}} \right)}{\log\left( \frac{\alpha}{\ell_{\nabla \psi_\alpha}} \right)}$ total inner loop*

*iterations where $\varepsilon > 0$. Then the outputs $(\boldsymbol{a}^{(t)}, \boldsymbol{b}^{(t)})_{t=0}^{T}$ satisfy $\min_{t=0,\ldots,T_{\boldsymbol{a}}-1} \left\| G_{\eta_{\boldsymbol{a}}}^{\varphi_\alpha}(\boldsymbol{a}^{(t)}) \right\|_2^2 \leq$*

$$\dfrac{1}{\frac{1}{\eta_{\boldsymbol{a}}} - \frac{\ell_{\nabla \psi_\alpha} + \frac{\ell_{\nabla \psi_\alpha}^2}{\alpha}}{2}} \left( \dfrac{\varphi_\alpha(\boldsymbol{a}^{(0)})}{T_{\boldsymbol{a}}} + \varepsilon \left( \left( \ell_{\nabla \psi_\alpha} + \frac{\ell_{\nabla \psi_\alpha}^2}{\alpha} + \varepsilon \right) \left( \frac{\ell_{\nabla \psi_\alpha} + \frac{\ell_{\nabla \psi_\alpha}^2}{\alpha}}{2} - \frac{1}{\eta_{\boldsymbol{a}}} \right) + \ell_{\psi_\alpha} \right) \right) .$$

*Proof of Theorem 4.2.* By Lemma B.4, we have:

$$\varphi_\alpha(\boldsymbol{a}^{(t+1)}) - \varphi_\alpha(\boldsymbol{a}^{(t)}) \leq \left( \frac{\ell_{\nabla \psi_\alpha} + \frac{\ell_{\nabla \psi_\alpha}^2}{\alpha}}{2} - \frac{1}{\eta_{\boldsymbol{a}}} \right) \left\| G_{\eta_{\boldsymbol{a}}}^{\varphi_\alpha}(\boldsymbol{a}^{(t)}) \right\|_2^2 + \left( (2\ell_{\varphi_\alpha} + \varepsilon) \left( \frac{\ell_{\nabla \psi_\alpha} + \frac{\ell_{\nabla \psi_\alpha}^2}{\alpha}}{2} - \frac{1}{\eta_{\boldsymbol{a}}} \right) + \ell_{\psi_\alpha} \right) \varepsilon$$

$$\tag{74}$$

Summing up the inequalities for $t = 0, \ldots, T_{\boldsymbol{a}} - 1$:

$$\varphi_\alpha(\boldsymbol{a}^{(T_{\boldsymbol{a}})}) - \varphi_\alpha(\boldsymbol{a}^{(0)}) \tag{75}$$

$$\leq \left( \frac{\ell_{\nabla \psi_\alpha} + \frac{\ell_{\nabla \psi_\alpha}^2}{\alpha}}{2} - \frac{1}{\eta_{\boldsymbol{a}}} \right) \sum_{t=0}^{T_{\boldsymbol{a}}-1} \left\| G_{\eta_{\boldsymbol{a}}}^{\varphi_\alpha}(\boldsymbol{a}^{(t)}) \right\|_2^2 + T_{\boldsymbol{a}} \left( (2\ell_{\varphi_\alpha} + \varepsilon) \left( \frac{\ell_{\nabla \psi_\alpha} + \frac{\ell_{\nabla \psi_\alpha}^2}{\alpha}}{2} - \frac{1}{\eta_{\boldsymbol{a}}} \right) + \ell_{\psi_\alpha} \right) \varepsilon$$

$$\tag{76}$$

After re-organizing, we obtain:

$$-\left(\frac{\ell_{\nabla\psi_\alpha} + \frac{\ell_{\nabla\psi_\alpha}^2}{\alpha}}{2} - \frac{1}{\eta_{\boldsymbol{a}}}\right)\sum_{t=0}^{T_{\boldsymbol{a}}-1}\left\|G_{\eta_{\boldsymbol{a}}}^{\varphi_\alpha}(\boldsymbol{a}^{(t)})\right\|_2^2 \tag{77}$$

$$\leq \varphi_\alpha(\boldsymbol{a}^{(0)}) - \varphi_\alpha(\boldsymbol{a}^{(T_{\boldsymbol{a}})}) + T_{\boldsymbol{a}}\left((2\ell_{\varphi_\alpha} + \varepsilon)\left(\frac{\ell_{\nabla\psi_\alpha} + \frac{\ell_{\nabla\psi_\alpha}^2}{\alpha}}{2} - \frac{1}{\eta_{\boldsymbol{a}}}\right) + \ell_{\psi_\alpha}\right)\varepsilon \tag{78}$$

$$\leq \varphi_\alpha(\boldsymbol{a}^{(0)}) + T_{\boldsymbol{a}}\left((2\ell_{\varphi_\alpha} + \varepsilon)\left(\frac{\ell_{\nabla\psi_\alpha} + \frac{\ell_{\nabla\psi_\alpha}^2}{\alpha}}{2} - \frac{1}{\eta_{\boldsymbol{a}}}\right) + \ell_{\psi_\alpha}\right)\varepsilon \tag{79}$$

Since $\frac{\ell_{\nabla\psi_\alpha} + \frac{\ell_{\nabla\psi_\alpha}^2}{\alpha}}{2} - \frac{1}{\eta_{\boldsymbol{a}}} < 0$ by the assumptions of our theorem, we can divide by $-\left(\frac{\ell_{\nabla\psi_\alpha} + \frac{\ell_{\nabla\psi_\alpha}^2}{\alpha}}{2} - \eta_{\boldsymbol{a}}\right)$ on both sides, and then take the minimum of $\left\|G_{\eta_{\boldsymbol{a}}}^{\varphi_\alpha}(\boldsymbol{a}^{(t)})\right\|_2^2$ across all $t \in [T_{\boldsymbol{a}}]$, to obtain:

$$T_{\boldsymbol{a}}\left(\min_{t=0,\ldots,T_{\boldsymbol{a}}-1}\left\|G_{\eta_{\boldsymbol{a}}}^{\varphi_\alpha}(\boldsymbol{a}^{(t)})\right\|_2^2\right) \tag{80}$$

$$\leq \frac{1}{\eta_{\boldsymbol{a}} - \frac{\ell_{\nabla\psi_\alpha} + \frac{\ell_{\nabla\psi_\alpha}^2}{\alpha}}{2}}\left(\varphi_\alpha(\boldsymbol{a}^{(0)}) + T_{\boldsymbol{a}}\left((2\ell_{\varphi_\alpha} + \varepsilon)\left(\frac{\ell_{\nabla\psi_\alpha} + \frac{\ell_{\nabla\psi_\alpha}^2}{\alpha}}{2} - \frac{1}{\eta_{\boldsymbol{a}}}\right) + \ell_{\psi_\alpha}\right)\varepsilon\right) \tag{81}$$

Dividing by $T_{\boldsymbol{a}}$ on both sides, we obtain:

$$\min_{t=0,\ldots,T_{\boldsymbol{a}}-1}\left\|G_{\eta_{\boldsymbol{a}}}^{\varphi_\alpha}(\boldsymbol{a}^{(t)})\right\|_2^2 \tag{82}$$

$$\leq \frac{1}{\frac{1}{\eta_{\boldsymbol{a}}} - \frac{\ell_{\nabla\psi_\alpha} + \frac{\ell_{\nabla\psi_\alpha}^2}{\alpha}}{2}}\left(\frac{\varphi_\alpha(\boldsymbol{a}^{(0)})}{T_{\boldsymbol{a}}} + \varepsilon\left((2\ell_{\varphi_\alpha} + \varepsilon)\left(\frac{\ell_{\nabla\psi_\alpha} + \frac{\ell_{\nabla\psi_\alpha}^2}{\alpha}}{2} - \frac{1}{\eta_{\boldsymbol{a}}}\right) + \ell_{\psi_\alpha}\right)\right) \tag{83}$$

$\square$

**Theorem 4.3** (PL Exploitability Convergence). *Suppose ADA is run on a PL-pseudo-game $\mathcal{G}$ which satisfies Assumption* 3.3 *with learning rates $\eta_{\boldsymbol{a}} \in \left[\ell_{\nabla\psi_\alpha} + \frac{\ell_{\nabla\psi_\alpha}^2}{\alpha}, \ell_{\nabla\psi_\alpha} + \frac{\ell_{\nabla\psi_\alpha}^2}{\alpha} + \frac{1}{2\mu}\right]$ and $\eta_{\boldsymbol{b}} = 1/\ell_{\nabla\psi_\alpha}$, for any number of outer loop iterations $T_{\boldsymbol{a}} \in \mathbb{N}_{++}$ and for $T_{\boldsymbol{b}} \geq \frac{\log\left(\frac{\varepsilon}{\nabla\psi_\alpha}\sqrt{\frac{2\alpha}{c}}\right)}{\log\left(\frac{\alpha}{\nabla\psi_\alpha}\right)}$ total inner loop iterations. Then the outputs $(\boldsymbol{a}^{(t)}, \boldsymbol{b}^{(t)})_{t=0}^T$ satisfy $\varphi_\alpha(\boldsymbol{a}^{(T_{\boldsymbol{a}})}) \leq \left[1 + 2\mu\left(\frac{\ell_{\nabla\psi_\alpha} + \frac{\ell_{\nabla\psi_\alpha}^2}{\alpha}}{2} - \frac{1}{\eta_{\boldsymbol{a}}}\right)\right]^{T_{\boldsymbol{a}}}\varphi_\alpha(\boldsymbol{a}^{(0)}) +$*

*$\left(\left(\ell_{\nabla\psi_\alpha} + \frac{\ell_{\nabla\psi_\alpha}^2}{\alpha} + \varepsilon\right)\left(\frac{\ell_{\nabla\psi_\alpha} + \frac{\ell_{\nabla\psi_\alpha}^2}{\alpha}}{2} - \frac{1}{\eta_{\boldsymbol{a}}}\right) + \ell_{\psi_\alpha}\right)\varepsilon.$*

*Proof of Theorem* 4.3. By Lemma B.4, we have:

$$\varphi_\alpha(\boldsymbol{a}^{(t+1)}) - \varphi_\alpha(\boldsymbol{a}^{(t)}) \tag{84}$$

$$\leq \left( \frac{\ell_{\nabla\psi_\alpha} + \frac{\ell_{\nabla\psi_\alpha}^2}{\alpha}}{2} - \frac{1}{\eta_{\boldsymbol{a}}} \right) \left\| G_{\eta_{\boldsymbol{a}}}^{\varphi_\alpha}(\boldsymbol{a}^{(t)}) \right\|_2^2 + \left( (2\ell_{\varphi_\alpha} + \varepsilon) \left( \frac{\ell_{\nabla\psi_\alpha} + \frac{\ell_{\nabla\psi_\alpha}^2}{\alpha}}{2} - \frac{1}{\eta_{\boldsymbol{a}}} \right) + \ell_{\psi_\alpha} \right) \varepsilon \tag{85}$$

$$\leq 2\mu \left( \frac{\ell_{\nabla\psi_\alpha} + \frac{\ell_{\nabla\psi_\alpha}^2}{\alpha}}{2} - \frac{1}{\eta_{\boldsymbol{a}}} \right) \left( \varphi_\alpha(\boldsymbol{a}^{(t)}) - \min_{\boldsymbol{a}\in\mathcal{X}} \varphi_\alpha(\boldsymbol{a}) \right) + \left( (2\ell_{\varphi_\alpha} + \varepsilon) \left( \frac{\ell_{\nabla\psi_\alpha} + \frac{\ell_{\nabla\psi_\alpha}^2}{\alpha}}{2} - \frac{1}{\eta_{\boldsymbol{a}}} \right) + \ell_{\psi_\alpha} \right) \varepsilon \tag{86}$$

$$\leq 2\mu \left( \frac{\ell_{\nabla\psi_\alpha} + \frac{\ell_{\nabla\psi_\alpha}^2}{\alpha}}{2} - \frac{1}{\eta_{\boldsymbol{a}}} \right) \varphi_\alpha(\boldsymbol{a}^{(t)}) + \left( (2\ell_{\varphi_\alpha} + \varepsilon) \left( \frac{\ell_{\nabla\psi_\alpha} + \frac{\ell_{\nabla\psi_\alpha}^2}{\alpha}}{2} - \frac{1}{\eta_{\boldsymbol{a}}} \right) + \ell_{\psi_\alpha} \right) \varepsilon \tag{87}$$

where the penultimate line is obtained from the projected-PL property combined with the fact that $\left( \frac{\ell_{\nabla\psi_\alpha} + \frac{\ell_{\nabla\psi_\alpha}^2}{\alpha}}{2} - \frac{1}{\eta_{\boldsymbol{a}}} \right) \leq 0$ by the theorem's assumptions. The last line was obtained from the fact that $\min_{\boldsymbol{a}\in\mathcal{X}} \varphi_\alpha(\boldsymbol{a}) = 0$. Re-organizing expressions:

$$\varphi_\alpha(\boldsymbol{a}^{(t+1)}) \leq \varphi_\alpha(\boldsymbol{a}^{(t)}) + 2\mu \left( \frac{\ell_{\nabla\psi_\alpha} + \frac{\ell_{\nabla\psi_\alpha}^2}{\alpha}}{2} - \frac{1}{\eta_{\boldsymbol{a}}} \right) \varphi_\alpha(\boldsymbol{a}^{(t)}) + \left( (2\ell_{\varphi_\alpha} + \varepsilon) \left( \frac{\ell_{\nabla\psi_\alpha} + \frac{\ell_{\nabla\psi_\alpha}^2}{\alpha}}{2} - \frac{1}{\eta_{\boldsymbol{a}}} \right) + \ell_{\psi_\alpha} \right) \varepsilon \tag{88}$$

$$\leq \left[ 1 + 2\mu \left( \frac{\ell_{\nabla\psi_\alpha} + \frac{\ell_{\nabla\psi_\alpha}^2}{\alpha}}{2} - \frac{1}{\eta_{\boldsymbol{a}}} \right) \right] \varphi_\alpha(\boldsymbol{a}^{(t)}) + \left( (2\ell_{\varphi_\alpha} + \varepsilon) \left( \frac{\ell_{\nabla\psi_\alpha} + \frac{\ell_{\nabla\psi_\alpha}^2}{\alpha}}{2} - \frac{1}{\eta_{\boldsymbol{a}}} \right) + \ell_{\psi_\alpha} \right) \varepsilon \tag{89}$$

Telescoping the sum for $t = 1, \ldots, T_{\boldsymbol{a}} - 1$, since $\left\| G_{\eta_{\boldsymbol{a}}}^{\varphi_\alpha} \right\|_2 \leq \left\| \nabla_{\boldsymbol{a}} \varphi_\alpha(\boldsymbol{a}) \right\|$ (Proposition 2.4, [116]), and $\left\| \nabla_{\boldsymbol{a}} \varphi_\alpha(\boldsymbol{a}) \right\| \leq \ell_{\nabla\psi_\alpha} + \frac{\ell_{\nabla\psi_\alpha}^2}{\alpha}$ by the Lipschitz-smoothness of $\varphi_\alpha$ (Theorem B.1), we obtain:

$$\varphi_\alpha(\boldsymbol{a}^{(T_{\boldsymbol{a}})}) \leq \left[ 1 + 2\mu \left( \frac{\ell_{\nabla\psi_\alpha} + \frac{\ell_{\nabla\psi_\alpha}^2}{\alpha}}{2} - \frac{1}{\eta_{\boldsymbol{a}}} \right) \right]^{T_{\boldsymbol{a}}} \varphi_\alpha(\boldsymbol{a}^{(0)}) + \left( (2\ell_{\varphi_\alpha} + \varepsilon) \left( \frac{\ell_{\nabla\psi_\alpha} + \frac{\ell_{\nabla\psi_\alpha}^2}{\alpha}}{2} - \frac{1}{\eta_{\boldsymbol{a}}} \right) + \ell_{\psi_\alpha} \right) \varepsilon \tag{90}$$

$\square$