# OpenReview forum: "Exploitability Minimization in Games and Beyond"
_NeurIPS.cc/2022/Conference — NeurIPS 2022 Accept_

### Official Review · Reviewer_PS8T · 2022-07-07

**Rating:** 6
**Confidence:** 3
**Soundness:** 3 good
**Presentation:** 2 fair
**Contribution:** 3 good

**Summary:**

This paper introduces two algorithms for finding an approximate variational equilbrium in pseudo-games with jointly convex constraints. Both algorithms look at setting up exploitability as a min-max problem with the regret-maxmizing strategy being an optimized variable, rather than a non-smooth max-regret objective. The authors provide theoretical bounds for the algorithms, and compare the performance to an existing algorithm on a three classes of random test problems.


**Questions:**

How is this work providing methods for exact computation of GNEs?  Both algorithms are iterative methods with bounds on solution quality after T iterations.

131: g>=0 iff g_i>=0 for all i?

132: d is some arbitrary, game-specific choice?  Is there intended to be some difference understood between describing it as g(x,y) = (g_1(x,y),...,g_d(x,y))T and g(x,y):XxY -> R^d similar to other functions?

135: min-max SE definition missing text describing leader (x) and follower (y) distinction?  Odd phrasing of definition, with the inequalities.  Those same inequalities make sense in the Goktas and Greenwald paper introducing min-max inequalities, where there are delta and epsilon approximations, but are non-intuitive choice here without the delta,epsilon. max_y f(x*,y) can not be greater than f(x*,y*).

144: is this the same d as on line 132?

150: "in which case X is simply a set"
Some other phrasing? X_i(a) is a common set across all i, but that's a subset of X -- as noted on 159.  Regardless of joint constraints, X(a) is also a set, as is the same {a | a in X(a)} set.  Saying "simply a set" seems to imply seem obvious reason to consider this set X, which is at least not obvious to this reader.

173, 178, 195, ...: Psi(b;a) is sometimes Psi(a,b), mostly later in the paper.  Psi(b;a) != Psi(a;b), so consistency here matters.

193: Key insight (also? rather?) is switch to looking at VEs instead of GNEs? Without the issue of a* being within X(a*), finding a VE with joint convex constraints seems to be a standard optimization problem. phi is non-smooth -- but with that switch, wouldn't something like Nesterov smoothing also work, with similar bounds as EDA?

204: What are those assumptions? Should also be included in the text, for contrast.

219: O(1/epsilon^(1/2)), rather than O(1/epsilon^2)?

249: How is b*=a a solution to the inner maximization? Is this max_{b in X} Psi(a,b)? For an arbitrary a, Psi(a,b=a)=0 seems unlikely to be a maximum.  The prior case where b=a* is in the argmax only seems to hold because Psi(a*,b) <= 0.

Algorithm 2: missing use of i in step 3?

Figure 1b: The scale required to contain the spikes in exploitability makes it hard to tell how close to 0 are things in the tails. Are the different methods notably different, or are they all inconsequentially close to 0?  Possibly solved by including a log-scale version of this plot instead / as well.

352: The paper states EDA has weaker assumptions on line 205, however here at line 353 AMPQP is expected to converge, while EDA is not. How do these two pieces fit together?


320: psuedo
330: we us -> we use a
349: methodsEDA
350: objectivesAMPQP


**Limitations:**

yes

**Strengths And Weaknesses:**

This work introduces two new algorithms EDA and ADA, with looser theoretical requirements than an existing algorithm, and the ability to find a stationary point in non-convex games, respectively.  On a set of randomized test problems, EDA is slightly worse and ADA is notably better, while both appear to be more likely to non-deterministically work than the baseline algorithm in situations where convergence isn't guaranteed.

I found the paper to be generally readable, but there were a number of sections that were less clear. One major piece that was less clear was exactly what the contributions of the paper were. Is it theoretical, reducing the bounds of AMPQP by a log factor -- although possibly working in a smaller set of games? Is it finding stationary points in a larger class of games -- why not test that?  Is it the empirical performance -- why introduce EDA then, when it is slightly outperformed on the test games chosen?


---- edited after author response
Clarifying the distinction between EDA and ADA,  EDA is a 1/epsilon method for finding a VE (and thus GNE) when regret is convex/concave, and ADA is a log(1/epsilon)/epsilon method for finding a stationary point as long as u is smooth and the feasible set is convex, or a VE if regret is invex.  This is described by lines 73-80.

---

> ### Author Response · Authors · 2022-08-01
> **Response to Reviewer PS8T**
>
> Thank you for your thorough review.
>
> For further explanation on our contributions and a comparison of our algorithms, please see the answer to all reviewers. We would like to note that introducing EDA is necessary, as it is theoretically optimal in the worst-case, while it seems difficult to prove a faster convergence rate for ADA; however, as ADA converges faster than EDA on randomly generated games, ADA’s average-case convergence rate may be faster than EDA’s.
>
> Our experiments test whether ADA converges to a stationary point of the exploitability in *arbitrary* bilinear pseudo-games with jointly convex constraints. We observe convergence to not only a stationary point but to an $\varepsilon$-GNE in all of the pseudo-games we simulate, even those for which ADA is not guaranteed to converge.
>
> Regarding your questions:
>
> We believe your question on exact computation refers to lines 61-66. We use the word exactly in that paragraph to mark the difference between convergence to an $\varepsilon$-GNE and a stationary point of the exploitability. This is because any stationary point of the exploitability is also an $\varepsilon$-GNE, where $\varepsilon$ is equal to the value of the exploitability at the stationary point (which although an approximate GNE, could be a bad approximation). Note that our experiments suggest that the approximation factor is in fact pretty good in practice.
>
> Line 131: Yes. This convention is used in the literature (see for instance [2]).
>
> Line 132: $d$ is an arbitrary game-specific choice. There is no difference between the two ways we write the function. The vector-valued function notation is sometimes preferred for compactness.
>
> Line 135: The left inequality in the Stackelberg equilibrium definition is necessary, i.e., $\max_{\mathbf{y} \in \mathcal{Y}} f(\mathbf{x}^*, y) \leq f(\mathbf{x}^*, \mathbf{y}^*)$, as otherwise one cannot guarantee that the $\mathbf{y}$-player best responds. That is, the right inequality $f(\mathbf{x}^*, \mathbf{y}^*) \leq \min_{\mathbf{x} \in \mathcal{X}} \max_{\mathbf{y} \in \mathcal{Y}} f(\mathbf{x}^*, y)$ can hold without $ \mathbf{y}^*$ being a best-response to the $\mathbf{x}$-player’s action. Without the left inequality, we obtain a robust strategy for the $\mathbf{x}$-player, but an arbitrary strategy for the $\mathbf{y}$-player.
>
> Line 144: Yes, this is the same $d$.
>
> Line 150: What we mean by “simply a set” is that the joint feasible action correspondence is a constant correspondence that has the same value at all action profiles.
>
> Lines 173, 178, 195: The notation in line 173 seems to be a typo. The correct version is as presented in line 178. Thank you for noticing!
>
> Line 193: The key insight is not how we solve the min-max optimization problem but rather our observation that exploitability minimization can be rephrased as a min-max optimization problem. Once this observation is made, one can use their desired method to solve this problem. EDA is an optimal algorithm to solve this problem, i.e., it achieves the lower bound of $O(\frac{1}{\varepsilon})$. It is not clear to us how Nesterov smoothing would be optimal, since extragradient descent ascent is the only known optimal method for saddle-point problems. Note also that smoothness is not an issue in section 3, as the cumulative regret is Lipschitz-smooth.
>
> Line 204: The assumption that the line is referring to is assumption 3.3. We will rearrange the text to make this clear.
>
> Line 219: We do actually mean $O(\frac{1}{\varepsilon^2})$. This is the convergence rate in last iterates for EDA, as shown by Golowich et al [1].
>
> Line 249: Our phrasing is confusing. What we mean is that there exists an $\mathcal{a}$ such that $\mathcal{b}^* = \mathcal{a}$. In particular, this $\mathcal{a}$ is exactly a GNE of the pseudo-game associated with cumulative regret. You can find a formal and more precise statement, as well as a proof, in Theorem C.1. of the appendix.
>
> Algorithm 2: This seems to be a typo. There should not be for loops over the players. We will fix this mistake. Thank you for noticing!
>
> Figure 1b: As we note in our text, in our experiments, the iterates generated by EDA and ADA converge to a GNE, implying that the exploitability of the resulting strategy profiles is inconsequentially small. We will consider using a log scale to present these results. Thank you for the suggestion!
>
> Line 352: The weaker assumptions, i.e., assumption 3.3, are the common set of assumptions made throughout the paper for the analysis of both EDA and ADA. However, the theorems for EDA hold only assuming convex-concave cumulative regret; as a result, the assumptions for EDA’s convergence are not weaker than those for AMPQP’s.
>
> [1] Golowich, Noah, et al. "Last iterate is slower than averaged iterate in smooth convex-concave saddle point problems." Conference on Learning Theory. PMLR, 2020
> [2] Facchinei, Francisco, and Christian Kanzow. "Generalized Nash equilibrium problems." Annals of Operations Research (2010)

---

> > ### Comment · Reviewer_PS8T · 2022-08-07
> > **Response to response**
> >
> > Thank you to the authors for the response.
> >
> > Along with some more reading -- especially von Hensinger and Kanzow -- I think the proposed contributions are now more clear. EDA is a 1/epsilon method for finding a VE (and thus GNE) when regret is convex/concave, and ADA is a log(1/epsilon)/epsilon method for finding a stationary point as long as u is smooth and the feasible set is convex, or a VE if regret is invex.  As described by lines 73-80.
> >
> >
> > Regarding Nesterov smoothing, I guess it's not applicable to the max phi problem, because u(x,y) is not necessarily xAy -- too familiar with games with matrix-like payoffs, too unfamiliar with pseudo-games. The thought was basically the regularized functions in von Heusinger and Kanzow. Given both ^V alpha and V alpha beta are smooth and convex, these would also seem to pair up with accelerated gradient descent for convergence to a VE at a rate of 1/epsilon , with different requirements (assumption 3.6 and 4.5).
> >
> > 135: Agree both sides are needed. The strangeness is the idea of max_y  f(x*,y) < f(x*,y*) when y* is in the domain of the max.
> >
> > 219: Thank you. Slower convergence with more properties didn’t make sense, I skimmed past last iterate.
> >
> > 352: understood, going back and forth I read 205 as an informal description of assumptions for EDA.  The phrase “Furthermore, these are weaker 205 assumptions than ones made to obtain the few known results on convergence rates to GNEs [78].” seems to be what triggered that – it might be worth stitching these three parts together in a different order, or with different phrasing: Assumption 3.3 is immediately followed by EDA with additional assumptions.

---

### Official Review · Reviewer_Xmxy · 2022-07-09

**Rating:** 5
**Confidence:** 2
**Soundness:** 2 fair
**Presentation:** 1 poor
**Contribution:** 2 fair

**Summary:**

The paper proposes methods for finding generalised Nash equilibria (GNE) for pseudo-games, which are a generalisation of classical games. They use a well known reformulation of the GNE problem as a min max problem. Then they show that a Extra-gradient method can be efficently used to find GNE in case in which the min max problem is convex-concave. Later that present a double loop gradient based algorithm to approximate the problem when the min-max reformulation is not convex concave. In this case the solution is a local Stackelberg equilibrium (stationary point of the minimal problem). They complement the theoretical analysis with an experimental evaluation

**Questions:**

Questions:
1) Are the results of Section 3 straightforward once one reformulates GNE as a VI problem and can any algorithm that solves VI be used here?
2) Is the main contribution of Section 4 the original reformulation of non-convex-non-concave pseudo games by means of the projected gradient operator? Can any algorithm that solves non-convex-non-concave min-max problem be used here?
3) Is the introduction of min max Stackelberg equilibrium necessary to the exposition?

Suggestions (minor):
3) Introduce early on an example of a pseudo game.
4) Define only jointly constrained pseudo-games.

See the Strengths And Weaknesses section for more detailed description of the questions.

**Strengths And Weaknesses:**

Form the point of view of the subject of study, the paper is original as not many papers in AI study pseudo-games. However, in the reviewer opinion, the technical contributions are not clear.
Section 3 seems straightforward as Th 5 of [57] says that any method adapt to solve Variations Inequalities can be employed to find GNE. The author should maybe clarify such point.
Section 4 seems more interesting but, again, the reformulation in terms of a min max problem seems to let us conclude that any algorithm that finds approximate stationary points of non-convex non concave min-max games can be used here and regularised exploitability was an already existing concept. However the the reformulation used here seems more involved as it needs to find the zeros the projected gradient operator. The authors should clarify such a point as it is central to the understanding of the contribution of the paper.

The paper is in general well written and well structured. However, in the reviewer’s opinion clarity could be slighly improved. I would suggest to include in the introduction a small toy example of a pseudo-game with jointly convex constraints.
The “min-max Stackelberg game” paragraph does not contribuite to the exposition of the paper. It’s almost never referenced later and I would suggest it to be moved in the appendix (maybe enlarged).
I would suggest to remove the general definition of a pseudo-game and just expose the case of joint constraint as the more general version is never actually used.

---

> ### Author Response · Authors · 2022-08-01
> **Response to Reviewer Xmxy**
>
> Thank you for your clarification questions. They are very useful!
>
> We would first like to clarify a point which the reference on line 184 might have obfuscated. To the best of our knowledge, the observation that exploitability minimization is a min-max optimization problem (Observation 3.1) is original to our paper. (For more details about this point, please refer to our rebuttal to all reviewers.)
>
> To answer your questions:
>
> 1. The results of section 3 are indeed straightforward once the problem is formulated as a min-max optimization problem, and in turn as a VI problem. Note that extragradient descent ascent is an optimal method, i.e., one which achieves the computational lower bound. Note also that there exists a VI formulation of variational equilibrium [1] for pseudo-games with jointly convex constraints, but this formulation is not as an exploitability-minimization method and applies only to monotone pseudo-games. The advantage of our method is that it extends convergence guarantees beyond monotone settings, e.g., to a class of bilinear pseudo-games.
>
> 2. The contribution of section 4 is 1) to reformulate any Lipschitz-smooth pseudo-game with joint constraints as a non-convex-concave min-max optimization problem with *Lipschitz-smooth exploitability*, and 2) to derive and analyze an efficient algorithm to solve the non-convex-concave optimization problem we pose using the projected gradient operator. Although any non-convex-concave min-max optimization algorithm can be used, all known algorithms are orders of magnitude slower than ours ($\tilde{O}(\frac{1}{\varepsilon^6})$ for the best-known algorithm [2] vs. $\tilde{O}(\frac{1}{\varepsilon})$ for ours). This is because our ADA algorithm exploits the properties of cumulative regret to obtain a Lipschitz-smooth exploitability function. More specifically, the algorithms in the literature thus far do not focus on the properties of the inner optimization problem, rather they focus on properties of the objective function. The property we identify as necessary for convergence is that the inner optimization problem be Lipschitz-smooth w.r.t. to the minimization variable.
>
> 3. The min-max Stackelberg game formalism is indeed inessential.
>
> We agree with you (and the other reviewers who were seeking more intuition) that an example would be helpful in the introduction.
>
> [1] Facchinei, Francisco, and Christian Kanzow. "Generalized Nash equilibrium problems." Annals of Operations Research 175.1 (2010): 177-211.
> [2] Thekumparampil, Kiran K., et al. "Efficient algorithms for smooth minimax optimization." Advances in Neural Information Processing Systems 32 (2019).

---

> > ### Comment · Reviewer_Xmxy · 2022-08-07
> > **Response to Response**
> >
> > I thank the authors for the clarifications. Now the contributions are more clear. I think in that these are not well presented in the current version of the paper.
> > However I'll raise my score accordingly.
> > Nonetheless, I think the paper needs to work on the exposition in order to make clearer such contributions, as all the reviewers were unsure about such contributions. I strongly encourage the authors to incorporate the discussions in the final version of the paper if accepted.

---

### Official Review · Reviewer_h6T8 · 2022-07-09

**Rating:** 6
**Confidence:** 2
**Soundness:** 3 good
**Presentation:** 2 fair
**Contribution:** 3 good

**Summary:**

This paper discusses Nash equilibrium computation in pseudo-games, which are games in which the players' legal strategy spaces may depend on other players' strategies. The paper reduces the problem to solving a certain min-max problem which is not convex-concave in the general case, and then gives algorithms for computing solutions of that min-max problem. The paper analyzes these algorithms both theoretically and practically, and obtain state of the art results in both.

**Questions:**

1. Is there any intuition for variational equilibrium? For example: why, in deviations, do we enforce the condition $\boldsymbol{a} \in \mathcal{X}$? That seems to imply to me that somehow, we concern ourselves with ways that *all of the players can collectively deviate* (i.e., the "joint deviation strategy" $\boldsymbol{a}$ must be feasible), yet at the same time we only care about whether each player individually profits from such a deviation when the others are playing equilibrium? This seems extremely weird to me coming from regular game land; the GNE definition seems more principled.
1. Why does the paper bother to define general pseudo-games, when all the results are seemingly restricted to pseudo-games with joint constraints? That seems like the more natural case anyway, and it simplifies a good amount of notation.
1. Along those lines, why is Section 4 titled "Pseudo-games with jointly convex constraints"? It seems like the entire paper devoted to pseudo-games with jointly convex constraints (by Assumption 3.3).
1. Does this paper give any new results for normal-form games, or are the results of this paper known for that particular case? It would be nice for such things to be explicitly discussed somewhere - perhaps the results can be framed as an extension of known normal-form results if that is true.


**Limitations:**

Yes

**Strengths And Weaknesses:**

The results in the paper are, to my knowledge, new and interesting, though I admit that I am not familiar with pseudo-games. Perhaps to aid readers like myself, I would advise the authors to include some extra intuition for people familiar with regular games but not pseudo-games. In the "Questions" section below, I include some questions that I had while reading the paper that would help greatly in that direction.

Along those lines, the paper generally seems a bit light on intuition. For example, the paper discusses the fact that algorithms are well known when the cumulative regret $\psi$ is convex-concave, and then introduces Assumption 3.3. How "far away" is Assumption 3.3 from convex-concavity? Is the analysis in the paper a straightforward extension of known analyses for convex-concave $\psi$, or do the weaker conditions introduce new technical problems to be resolved? If the latter, what are they?

---

> ### Author Response · Authors · 2022-08-01
> **Response to Reviewer h6T8**
>
> Thank you for your comments and questions!
>
> We would like to answer your questions, which we hope you will find clarifying.
>
> Assumption 3.3 is an assumption we make throughout the paper for both algorithms. It is necessary for the problem to be tractable. This assumption is standard in the literature for computing GNE in pseudo-games and Nash equilibria in games (for instance, see [1] for pseudo-games, or [2] for games). We note that on their own these assumptions do not provide polynomial-time computational guarantees to approximate GNEs, as otherwise the computation of GNE would be polynomial time in most pseudo-games of interest, which is unlikely due to the PPAD-completeness of the Nash equilibrium problem. Likewise, Assumption 3.3 does not imply convex-concavity of the cumulative regret. Our contribution is to show that under Assumption 3.3, convergence to a stationary point of the exploitability, i.e., to an $\varepsilon$-GNE, can be achieved.
>
> Once you notice, as we did (for further explanation, please refer to our rebuttal to all reviewers), that the exploitability-minimization problem can be restated as a min-max optimization problem, deriving and analyzing EDA is a straightforward extension of known results on convex-concave saddle point problems. ADA, however, is not a straightforward extension of known results, as our analysis required a novel Moreau envelope theorem, by which we achieve a convergence rate which is orders of magnitude faster than known algorithms for non-convex-concave min-max optimization problems ($\tilde{O}(\frac{1}{\varepsilon^6})$ for the best-known algorithm [2] vs. $\tilde{O}(\frac{1}{\varepsilon})$ for ours).
>
> Regarding your other questions:
>
> 1. At a GNE, we consider for each player *unilateral* deviations over *individually feasible strategy sets, given the GNE strategies of the opponents*. That is, a GNE does not allow new strategies to become available to a player’s opponents if the said player deviates.
>
>    At a variational equilibrium (VE), we consider *unilateral* deviations over *collectively feasible strategies*. That is, a VE *does* allow new strategies to become available to a players’ opponents if the said player deviates.
>
>    A VE is a refinement of a GNE, which is robust to the changes in the strategy sets of opponents once a player deviates. In our opinion, the robust aspect of VE makes them even more justifiable than GNE, since it seems unreasonable that players would not strategize about how their opponents’ feasible strategy sets would change once they deviate from their strategy.
>
> 2. and 3. Thank you for these suggestions. We will implement them.
>
> 4. All our results apply directly to normal-form games. In particular, to the best of our knowledge our EDA algorithm provides the broadest polynomial-time computation guarantees for Nash equilibrium in normal-form games, i.e., zero-sum, potential, and a large class of monotone and a class of bilinear pseudo-games. Our results also suggest that exploitability minimization is a one-size-fits-all method for computing Nash equilibrium in games, since EDA converges to a NE not only in games for which we have polynomial-time computation guarantees (i.e., zero-sum, potential, and monotone games) but also extends them to a class of bilinear pseudo-games and beyond. Additionally, this universal method is optimal; that is EDA’s asymptotic convergence rate is the same as the lower bound on convergence to Nash equilibrium in two-player zero-sum games. Finally, although our theory only guarantees that ADA converges to a stationary point of the exploitability in Lipschitz-smooth games, our experiments suggest that in practice ADA with proper initialization is effective at finding Nash equilibria (rather than mere stationary points) in some Lipschitz-smooth games, making ADA a strong candidate algorithm for computing Nash equilibrium in general-sum games.
>
> [1] Jordan, Michael I., Tianyi Lin, and Manolis Zampetakis. "First-Order Algorithms for Nonlinear Generalized Nash Equilibrium Problems." arXiv preprint arXiv:2204.03132 (2022).
>
> [2] Daskalakis, Constantinos, Stratis Skoulakis, and Manolis Zampetakis. "The complexity of constrained min-max optimization." Proceedings of the 53rd Annual ACM SIGACT Symposium on Theory of Computing. 2021.
>
> [3] Facchinei, Francisco, and Christian Kanzow. "Generalized Nash equilibrium problems." Annals of Operations Research 175.1 (2010): 177-211.

---

> > ### Comment · Reviewer_h6T8 · 2022-08-06
> > **Response to response**
> >
> > Thank you for the response. It has clarified several concerns I had in my review, and as such I raise my score. I would recommend that the authors incorporate these discussions into the final paper, as they would, in my opinion, greatly increase the readability of the paper.

---

### Official Review · Reviewer_y2D9 · 2022-07-11

**Rating:** 6
**Confidence:** 2
**Soundness:** 3 good
**Presentation:** 3 good
**Contribution:** 3 good

**Summary:**

The paper proposes two algorithms for finding approximate local Nash equilibria in pseudo-games. The algorithms are based on a new function, which roughly measures each player’s regret in some strategy profile a to a strategy in some other strategy profile b. A strategy profile a is an equilibrium if and only if it minimizes the maximum of regret over b to 0. The algorithms are then based on optimizing a and b with gradient ascent/descent. Various technical problems are addressed, e.g., gradient steps taking the solutions out of the feasible sets. The authors show that under convexity/concavity of the regret (and a few other assumptions), the algorithm finds (approximate) local Nash equilibria. They also demonstrate that they algorithms converge faster than existing algorithms in a few experiments.

**Questions:**

Most of the above questions seem relatively minor, but I’d still be interested in explanations.

**Limitations:**

Yes.

**Strengths And Weaknesses:**

This paper’s contribution seems good as far as I can tell. Unfortunately, I don’t know this literature (pseudo-games, finding equilibria) very well. So I’m not very confident in my judgment.

Some minor comments:

Superfluous apostrophe in line 140.

Line 148: Are we assuming differentiability?

Line 229: “More generally, cumulative regret can be a non-convex-concave function, in which case a minimax theorem does not hold, which precludes the existence of a NE.“
I found this sentence hard to understand. I suppose you are talking about the game induced by \Psi, not the original game?

The sentence in Line 238ff. is not grammatical, I believe.

Line 262: I assume one of the \psi_\alpha should be \phi_alpha.

Line 270: This is a weird indicator function, especially with the blackboard-bold 1, but okay.
Also, how is this indicator function continuous (or convex)? It looks rather discontinuous. Is this just on X?

I don’t understand Line 2 and 3 of Algorithm 2. Why is line 3 performed n times, where n is the number of players?

---

> ### Author Response · Authors · 2022-08-01
> **Response to Reviewer y2D9**
>
> Thank you for your review!
>
> We would like to first note that our algorithms converge to global GNEs in pseudo-games with jointly convex constraints and convex-concave cumulative regret. (We mention this point explicitly because the first sentence of your review suggests that our algorithms might converge to a local GNE in such pseudo-games). Pseudo-games with convex-concave cumulative regret not only cover a vast majority of pseudo-games for which we know how to compute GNE (zero-sum, potential, and monotone), but further include a class of bilinear pseudo-games. Our result not only provides one of the first polynomial-time computation guarantees for GNEs in such a broad class of pseudo-games, but also provides a novel way to compute Nash equilibria in polynomial time in a class of bilinear pseudo-games, for which polynomial-time computation likely does not hold in general. In our experiments, our algorithms also outperform known algorithms (which are arguably more complicated) for computing GNEs in pseudo-games [2].
>
> Regarding your questions:
>
> Line 148: One does not have to assume that the utilities are differentiable, and can instead replace the gradient of the utility function by an arbitrary subgradient.
>
> Line 229: Yes, we are referring to the min-max optimization problem induced by the function $\psi$, and by Nash equilibrium we mean a saddle point of the cumulative regret.
>
> Line 270: The definition of indicator function we use is standard in the convex optimization literature (see, for example, [1]). The indicator function being convex is a well known result (see Theorem A.1. of [1]), and can be shown as follows: Suppose  $C \subset \mathbb{R}^n$ is a convex set. Consider the case such that $\mathbf{x}, \mathbf{y} \in C$,  then $\mathbb{1}_C (\lambda \mathbf{x} + (1-\lambda) \mathbf{y}) = 0$ since the convex combination of elements in a convex set are once again in that set. Additionally, $\lambda \mathbb{1}_C (\mathbf{x}) + (1-\lambda) \mathbb{1}_C (\mathbf{y}) = \lambda (0) + (1-\lambda) (0) = 0$. Hence, we have $\mathbb{1}_C (\lambda \mathbf{x} + (1-\lambda) \mathbf{y}) \leq \lambda \mathbb{1}_C (\mathbf{x}) + (1-\lambda) \mathbb{1}_C (\mathbf{y})$, which proves that $\mathbb{1}_C$ is convex over C . The convexity of $\mathbb{1}_C$ over $\mathbb{R}^n$ follows similarly by considering cases where 1) $\mathbf{x} \in C, \mathbf{y} \notin C$, 2) $\mathbf{x} \notin C, \mathbf{y} \in C$, and 3) $\mathbf{x},  \mathbf{y} \notin C$. For instance suppose that  $\mathbf{x},  \mathbf{y} \notin C$, then $\lambda \mathbb{1}_C (\mathbf{x}) + (1-\lambda) \mathbb{1}_C (\mathbf{y}) = \infty$ and $\mathbb{1}_C (\lambda \mathbf{x} + (1-\lambda) \mathbf{y})$ is equal to 0 or $\infty$, which once again implies that $\mathbb{1}_C (\lambda \mathbf{x} + (1-\lambda) \mathbf{y}) \leq \lambda \mathbb{1}_C (\mathbf{x}) + (1-\lambda) \mathbb{1}_C (\mathbf{y})$.
>
> As for lines 2 and 3 in algorithm 2, they are typos. There should be no reference to the players at all. Thank you for noticing!
>
> [1] https://link.springer.com/content/pdf/bbm%3A978-3-662-52696-5%2F1.pdf
>
> [2] Jordan, Michael I., Tianyi Lin, and Manolis Zampetakis. "First-Order Algorithms for Nonlinear Generalized Nash Equilibrium Problems." arXiv preprint arXiv:2204.03132 (2022).

---

### Author Response · Authors · 2022-08-01
**Response to all Reviewers**



Thank you all for taking the time to review our paper!

All the papers in the literature thus far have formulated the exploitability-minimization problem as $\min_{\mathbf{a} \in \mathcal{X}(\mathbf{a})} \varphi(\mathbf{a})$, which has hindered progress in the computation of GNE, so much so, that Facchinei et al. in their foundational survey  mention that “$\varphi(\mathbf{a})$ is hard to compute in general, [as it is] usually non-differentiable” (see [1], top of page 189), making the formulation of the GNE problem as an exploitability-minimization problem (ironically) unexploitable!

Our main contribution is the observation that the exploitability-minimization problem can be recast as a simple and seemingly trivial min-max optimization problem (Observation 3.1). To the best of our knowledge we are the first to notice this equivalence. Some evidence in support of this claim is that otherwise the literature would have not have emphasized the arguably more complicated relaxation or Newton methods to overcome the nondifferentiability of exploitability.

*The strength of our paper lies in the simplicity of this previously unobserved min-max formulation, which allows us to provide an affirmative answer to Flam and Ruszczynski’s 20+-year-old conjecture [2] that projected gradient methods could be developed to minimize exploitability in pseudo-games.* We note that existing methods [1] make use of the exploitability concept to analyze algorithms that take alternative approaches, but they are not exploitability-minimization algorithms per se, as exploitability itself is not explicitly minimized.

More precisely, we solve for $\varepsilon$-GNE in convex-concave cumulative regret pseudo-games with jointly convex constraints (e.g., zero-sum, potential, Cournot, a large class of monotone pseudo-games and a novel class of bilinear pseudo-games, namely those with with convex second-order approximations (see Proposition (4c) of [2]) in $O(\frac{1}{\varepsilon})$ iterations (Algorithm 1 - EDA, Theorem 3.4). We also obtain convergence to an $\varepsilon$-stationary point of the exploitability (Algorithm 2 - ADA, Theorem 4.2) more generally in all Lipschitz-smooth pseudo-games with jointly convex constraints in $O(\frac{log(\varepsilon)}{\varepsilon})$ iterations. We note that ADA is also guaranteed to converge to $\varepsilon$-GNE in the class of convex-concave cumulative regret pseudo-games with jointly convex constraints at the same rate (Corollary of Theorem 4.2 and Theorem 4.3). These results, in our opinion, are very interesting, as combined with the $O(\frac{1}{\varepsilon})$ lower bound complexity of Nash equilibrium in two-player games [3], they reveal that the complexities of solving zero-sum, potential, a large class of monotone pseudo-games, and a class of bilinear pseudo-games are all the same as those of the corresponding *games*!

EDA is thus an optimal algorithm, which achieves the same convergence rate as AMPQP in pseudo-games in which the latter converges (i.e., monotone pseudo-games with jointly affine constraints). Further, its convergence guarantees extend beyond monotone settings to a class of bilinear pseudo-games for which AMPQP is not guaranteed to converge. Similarly ADA converges, albeit at a slightly slower rate, in the class of pseudo-games for which AMPQP is guaranteed to converge, and it more generally converges in all Lipschitz-smooth pseudo-games with jointly convex constraints to an $\varepsilon$-stationary point of the exploitability. Further, as we demonstrate experimentally, ADA can converge to an $\varepsilon$-GNE with suitable initialization. This makes ADA the first algorithm to solve for the $\varepsilon$-GNE (where $\varepsilon$ is equal to the value of the exploitability at the stationary point of the exploitability) of Lipschitz-smooth pseudo-games—and, as a result, the $\varepsilon$-Nash equilibria of Lipschitz-smooth games, for which, as far as we are aware, such broad convergence guarantees are not known. The main tool we use in proving the convergence of ADA is a novel parametric Moreau envelope theorem (Theorem 4.1), which is likely valuable as an optimization tool in other applications as well.

All in all, our results 1) improve upon the state-of-the-art by extending polynomial-time convergence guarantees to a class of pseudo-games beyond monotone, and 2) provide a general approach to approximate, with convergence guarantees, solutions to Lipschitz-smooth pseudo-games with jointly convex constraints, which, as far as we are aware, is the first result of its kind.

[1] Facchinei, Francisco, and Christian Kanzow. "Generalized Nash equilibrium problems." Annals of Operations Research 175.1 (2010)

[2] Sjur Flam and Andrzej Ruszczynski. Noncooperative Convex Games. International Institute for Applied Systems Analysis (1994)

[3] Thekumparampil, Kiran K., et al. "Efficient algorithms for smooth minimax optimization." Advances in Neural Information Processing Systems 32 (2019)

---

### Meta-Review · Area_Chair_xNXE · 2022-08-26

**Recommendation:** Accept
**Confidence:** Certain

**Metareview:**

On the one hand, this paper does not suffer from important criticisms. On the other hand, the paper has not a champion as all the Reviewers set their score between Borderline and Weak Accept.

The main weakness concerns the need to improve the presentation and clarify the contributions. I believe this issue can be addressed in a minor revision which does not require a further step of revision. Therefore, I don't see crucial reasons to reject the paper.

**Award:**

No

---

### Decision · Program_Chairs · 2022-09-14

Accept